# Optogenetic activation of dorsal raphe serotonin neurons induces brain-wide activation

Hiro Taiyo Hamada [1,2] ✉, Yoshifumi Abe[3], Norio Takata [3], Masakazu Taira[1], Kenji F. Tanaka [3] & Kenji Doya [1] ✉

Serotonin is a neuromodulator that affects multiple behavioral and cognitive functions. Nonetheless, how serotonin causes such a variety of effects via brain-wide projections and various receptors remains unclear. Here we measured brain-wide responses to optogenetic stimulation of serotonin neurons in the dorsal raphe nucleus (DRN) of the male mouse brain using functional MRI with an 11.7 T scanner and a cryoprobe. Transient activation of DRN serotonin neurons caused brain-wide activation, including the medial prefrontal cortex, the striatum, and the ventral tegmental area. The same stimulation under anesthesia with isoflurane decreased brain-wide activation, including the hippocampal complex. These brain-wide response patterns can be explained by DRN serotonergic projection topography and serotonin receptor expression profiles, with enhanced weights on 5-HT1 receptors. Together, these results provide insight into the DR serotonergic system, which is consistent with recent discoveries of its functions in adaptive behaviors.

Serotonin, 5-hydroxytryptamine (5-HT), serves crucial behavioral and cognitive functions, including regulation of the awake-sleep cycle, mood, reward, and punishment predictions[1,2], waiting behaviors[3,4], and learning[5]. Serotonin is also a major target of medications for psychiatric disorders, notably major depression, by selective serotonin re-uptake inhibitors (SSRIs). The dorsal raphe nucleus (DRN) is a major source of serotonergic projections to brain-wide targets, including the frontal cortex, the basal ganglia, the ventral tegmental area (VTA), and other mid-brain regions[6]. The DRN contains glutamatergic, GABAergic, and dopaminergic neurons, in addition to serotonergic neurons[7]. Optogenetic manipulation of activities of serotonin neurons has revealed their functions in reward[8,9], waiting for delayed rewards[3,4], anxiety[10,11], and learning[5,12]. Nonetheless, associations between brain-wide serotonergic projections and brain responses remain unclear[13,14].

In order to clarify the global picture of serotonergic control of brain activities, we performed optogenetic functional MRI experiments using Tph2-ChR2(C128S) transgenic mice and an 11.7 T MRI scanner with a cryoprobe for high S/N measurements. A previous optogenetic fMRI experiment activated DRN serotonin neurons using the Pet1 promoter under anesthesia and showed widespread suppression of cortical activities[15]. We introduced an acclimation protocol to allow stable fMRI recording in awake mice[16–18] and found that DRN serotonin activation in an awake state induced increased brain-wide activity in both cortical and sub-cortical areas. In contrast, under anesthesia with isoflurane, DRN serotonin activation decreased activities in those areas, consistent with a previous study[15].

We further found that regional BOLD responses can be approximated by a linear model with projection densities of DRN serotonin neurons and expression profiles of different types of serotonin receptors, with negative coefficients for 5-HT1 receptors and positive coefficients for 5-HT2 receptors. These results suggest that serotonergic neurons activate brain-wide regions, and they demand reconsideration of the implications of results under anesthesia.

[1]Neural Computation Unit, Okinawa Institute of Science and Technology Graduate University, Okinawa, Japan. [2]Research & Development Department, Araya Inc, Tokyo, Japan. [3]Division of Brain Sciences, Institute for Advanced Medical Research, Keio University School of Medicine, Tokyo, Japan. ✉e-mail: hamada_h@araya.org; doya@oist.jp

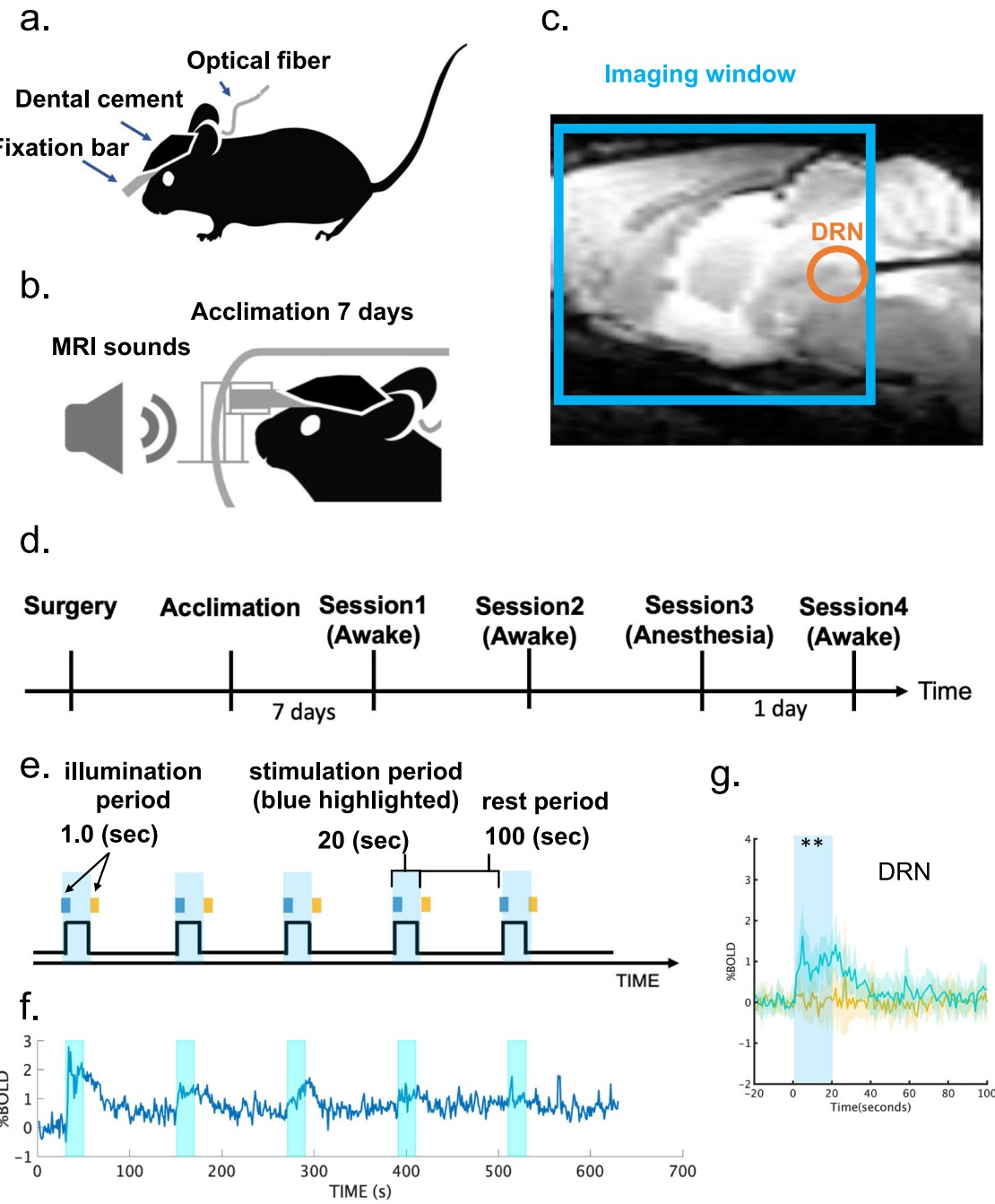

**Fig. 1 | Experimental setup for opto-fMRI. a** Surgical setup. A fixation bar was mounted on a mouse's head with dental cement and an optical cannula was inserted into the dorsal raphe nucleus (DRN) **b** Acclimation protocol. A mouse was exposed daily to recorded MRI scanning sounds for 2 h for 7 days prior to fMRI sessions. **c** Location of an optical cannula. An optical cannula was inserted horizontally into the DRN from the cerebellum. A cyan window indicates an imaging window for T2* sessions. **d** Experimental schedule. After surgery and 7 days acclimation, the first fMRI recording session was performed. A replicate session was also performed a few weeks later. Finally, experiments for anesthesia effects were performed. On the 1st day, an fMRI recording with general anesthesia using isoflurane 1% was performed, and an fMRI recording without anesthesia was conducted thereafter **e** Stimulation protocol. In an fMRI session, optogenetic stimulation was initiated 90 s after an imaging session began. Blue illumination (1.0 s; 473 nm) was applied at the onset of stimulation, and yellow illumination (1.0 s; 593 nm) followed the offset of stimulation 20 s after blue illumination. **f** Average time course of % BOLD signals in DRN. Time series of % BOLD signals were individually extracted from the DRN in Sessions 1 and 2, and time series were averaged across transgenic mice ($n = 8$). Each cyan window indicates the stimulation period during a session for 20 s. **g** Alignment of average % BOLD responses in the DRN. % BOLD responses were aligned at the time of blue illumination (1.0 s) in session 1. Blue and yellow lines indicate % BOLD responses in the DRN from blue and yellow stimulation sessions from the transgenic group ($n = 8$), respectively. Blue and yellow highlights indicate the range of mean values ± standard deviation (SD). A cyan window indicates stimulation duration during a session for 20 s. ** indicates $p_{FDR\text{-}corrected} < 0.01$ with two-tailed $t$-tests, FDR corrected.

## Results

In order to clarify effects of brain-wide serotonergic projections, we employed optogenetic fMRI, combining optogenetic stimulation of serotonergic neurons in the dorsal raphe nucleus (DRN) and measurements of whole-brain activities using functional MRI. We used Tph2-ChR2(C128S) transgenic mice[11,19] which selectively express step-type channel rhodopsin in serotonin neurons[4,11]. We used an 11.7 T MRI scanner (Bruker BioSpec 117/11) with a cryoprobe for high S/N recording of BOLD signals from the whole brain, except the olfactory bulb and the posterior cerebellum.

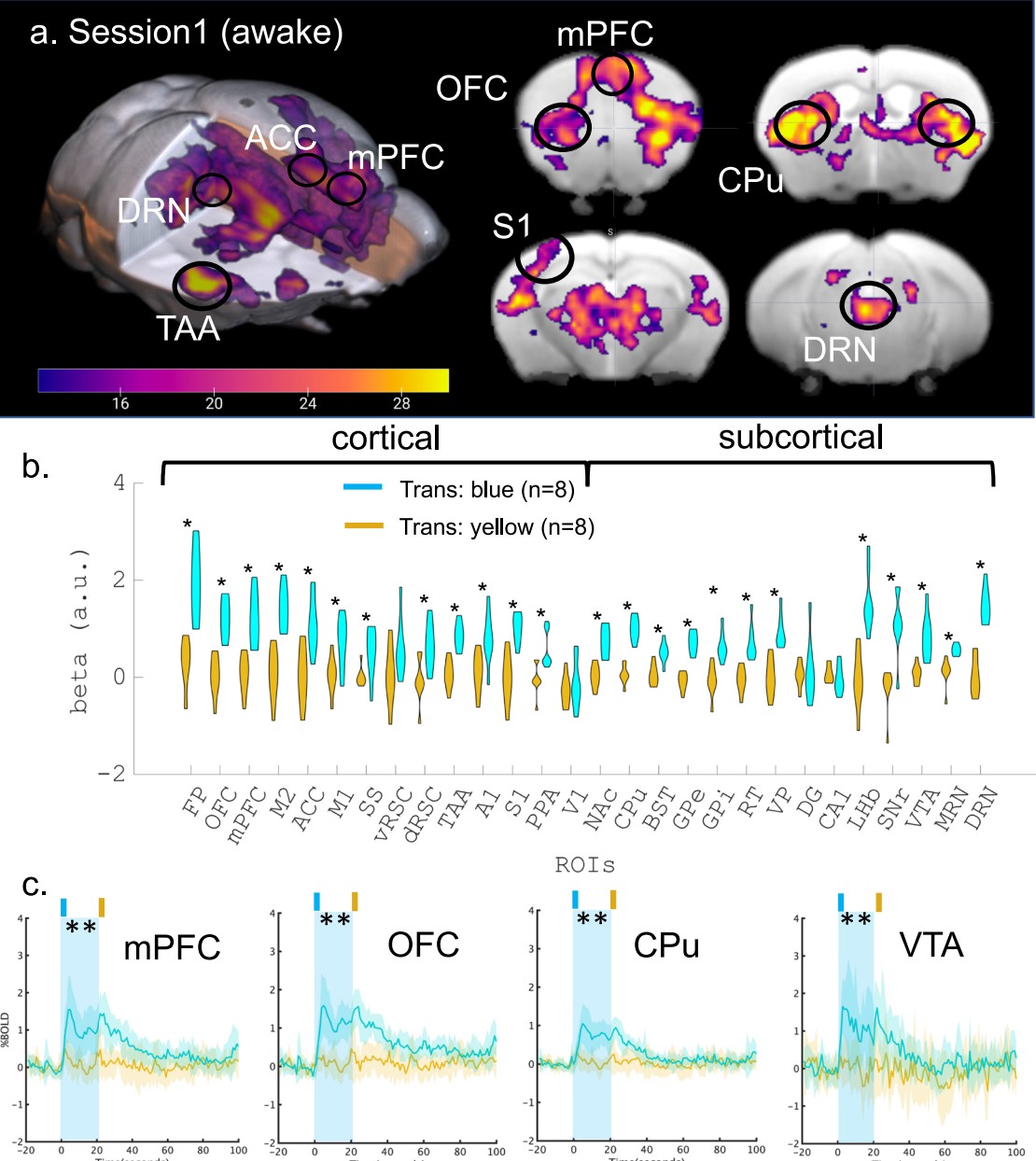

**Fig. 2 | BOLD response to optogenetic stimulation of serotonin neurons in the DRN. a** Functional map of optogenetic illumination. A functional map was calculated by contrasting BOLD responses between blue and yellow illumination in the transgenic ($n = 8$) group ($p_{FWE-corrected} < 0.05$, two-tailed paired $t$-tests and cluster-corrected controlling family-wise error (FWE) with probabilistic threshold-free cluster enhancement (pTFCE, see "General Linear Model (GLM) for BOLD signals" section)). Magenta color indicates the $t$-value. **b** Comparison of beta values in multiple ROIs. Beta values were obtained from GLM analysis in Session 1, and were extracted and averaged based on ROIs from the Allen Brain Atlas (Table 1). Differences of beta values between blue and yellow illumination sessions were contrasted in the transgenic group. * indicates $p_{FDR-corrected} < 0.05$ (Two-tailed paired $t$-tests were performed, and multiple comparison was corrected with false discovery rate (FDR)). Cyan and magenta colors indicate blue and yellow illumination, respectively. **c** Alignment of average % BOLD responses in the mPFc, OFC, CPu, and VTA. % BOLD responses were aligned at the time of blue illumination (1.0 s) in Session 1. Blue illumination (1.0 s; 473 nm) was applied at the onset of stimulation, and yellow illumination (1.0 s; 593 nm) followed 20 s the offset of the stimulation after blue illumination. A cyan window indicates stimulation duration during a session for 20 s. Blue and yellow lines indicate % BOLD responses in the mPFc, OFC, CPu, and VTA from blue and yellow illumination from transgenic groups ($n = 8$), respectively. Blue and yellow highlights indicate the range of mean values ± SD. A cyan window indicates stimulation duration during a session for 20 s. ** indicates $p_{FDR-corrected} < 0.01$ with two-tailed $t$-tests, FDR corrected.

Prior to experiments, the transgenic and wild-type (WT) groups (Transgenic group: $n = 8$; Wild-type: $n = 6$) underwent surgery to attach a plastic fixation bar and implant an optical cannula in the DRN. Acclimation training in a mock chamber followed (Fig. 1. See "Methods" for details). After fMRI experiments, animals were tested in a reward waiting task[3,4] to confirm the effectiveness of optogenetic stimulation (Supplementary Fig. S1).

## DRN serotonin activation induces a brain-wide increase in BOLD signals

We measured BOLD responses under optogenetic stimulation of DRN serotonin neurons in an ON/OFF block design protocol (Fig. 1e). Photo-stimulation increased BOLD signals in the DRN (Fig. 1f), with a tendency for reduced responses after repeated stimulation. On average, over five stimulations, blue light stimulation induced a

**Table 1 | Regions of interest (ROIs) and their abbreviations**

| ROI id | Area | ROI name | Abbreviations |
|---|---|---|---|
| 1 | Cortical regions | Frontal polar area | FP |
| 2 | | Orbital area | OFC |
| 3 | | Medial prefrontal cortex | mPFC |
| 4 | | Secondary motor area | M2 |
| 5 | | Anterior cingulate cortex | ACC |
| 6 | | Primary motor area | M1 |
| 7 | | Supplementary somatosensory area | SS |
| 8 | | Ventral retrosplenial cortex | vRSC |
| 9 | | Dorsal retrosplenial cortex | dRSC |
| 10 | | Temporal association area | TAA |
| 11 | | Primary auditory area | A1 |
| 12 | | Primary somatosensory area | S1 |
| 13 | | Posterior parietal association area | PPA |
| 14 | | Primary visual area | V1 |
| 15 | Sub-cortical regions | Nucleus accumbens | NAc |
| 16 | | Caudate putamen | CPu |
| 17 | | Bed nucleus of steria terminalis | BST |
| 18 | | Globus pallidus external part | GPe |
| 19 | | Globus pallidus internal part | GPi |
| 20 | | Thalamic reticular nucleus | RT |
| 21 | | Ventral pallidus | VP |
| 22 | | Dentate gyrus of hippocampus | DG |
| 23 | | CA1 of hippocampus | CA1 |
| 24 | | Lateral habenula | LHb |
| 25 | | Substantia nigra pars reticulata | SNr |
| 26 | | Ventral tegmental area | VTA |
| 27 | | Medial raphe nucleus | MRN |
| 28 | | Dorsal raphe nucleus | DRN |

ROIs are ordered from anterior to posterior for cortical and sub-cortical areas.

significantly higher BOLD response than did yellow-light stimulation ($p_{FDR-corrected} < 0.05$, correction of multiple comparisons with a false discovery rate (FDR), see "General Linear Model (GLM) for BOLD signals" section; Fig. 1g and Supplementary Fig. S5).

A group analysis with a general linear model (GLM) of photo-activated BOLD responses between blue and yellow stimulation from the transgenic group ($n = 8$) revealed a brain-wide increase in BOLD responses, significantly in the frontal cortex, the caudate putamen (CPu), temporal cortical regions, the thalamic complex, and the VTA ($p_{FWE-corrected} < 0.05$, cluster-wise correction controlling family-wise error (FWE) with probabilistic threshold-free cluster enhancement (pTFCE); Fig. 2a). Trial-wise BOLD responses also showed consistent brain-wide activation by blue, but not yellow stimulation ($p_{uncorrected} < 0.05$, Supplementary Fig. S2a, b).

We next selected 28 regions of interest (ROIs) from cortical and sub-cortical areas that are involved in valence processing, according to the Allen mouse brain atlas (Table 1, see "Regions of interest" section). We selected components of basal ganglia such as the caudate putamen (CPu), nucleus accumbens (NAc), and globus pallidus (GP)[20]. Further-more, other sub-cortical regions, including the bed nucleus of the stria terminalis (BST)[21], the hippocampus[22], and the lateral habenula (LHb)[23] were also selected. Then, we extracted regional average beta values (regression coefficients) from the GLM. Comparison of regional beta values under blue and yellow stimulation from the transgenic group revealed that BOLD responses were significantly higher in blue

illumination sessions in 12 of 14 cortical regions, including the OFC, mPFC, and ACC, and 12 of 14 sub-cortical regions, including the CPu, NAc, GP, VTA, LHb, and DRN (Fig. 2b) although no statistical differences were found in the WT group (Supplementary Fig. S4). Moreover, there were no significantly higher BOLD responses in vRSC, V1, DG, or CA1.

The week after Session 1, we performed another session under the same stimulation parameters as in Session 1. Patterns of BOLD responses were similar to those in Session 1 (transgenic: $n = 7$, WT: $n = 6$, Supplementary Fig. S3). Pearson's correlation of beta values in Sessions 1 and 2 was $r = 0.875$, which was statistically significant ($p_{uncorrected} < 1e^{-8}$; Supplementary Fig. S7.a). We further checked regional differences of responses in peak amplitude and peak timing by extracting regional BOLD signals (Supplementary Fig. S5-9). Peak amplitude and peak timing of BOLD responses varied across the 28 ROIs (Supplementary Fig. S8.c, d), with the largest responses in the VTA, SNr, and FP. Both amplitude and timing of BOLD responses in Sessions 1 and 2 were significantly correlated under blue light stimulation (amplitude: $r = 0.886$, $p_{uncorrected} < 1e^{-8}$; timing: $r = 0.679$, $p_{uncorrected} < 0.005$), but not yellow stimulation (amplitude: $r = -0.06$, $p_{uncorrected} = 0.766$; $r = 0.037$, $p_{uncorrected} = 0.855$; Supplementary Fig. S8.a, b). These results indicate that our photo-stimulation of DRN induced consistent activation of DRN serotonergic neurons, which caused distinct spatial and temporal responses in multiple cortical and sub-cortical regions.

## DRN serotonin activation induces a brain-wide negative response in BOLD signals under anesthesia

Contrary to our above results, a previous optogenetic fMRI study reported that DR serotonin activation evoked negative cortical cerebral blood volume (CBV) responses, an enhanced proxy of BOLD signals, from the entire cortical region under anesthesia[15]. There are two critical differences between that study and that one, optogenetic probes and anesthesia. The previous study used the ePET-Cre mouse line and AAV harboring Cre-dependent ChR2 and applied 20-Hz photo-stimulation. In contrast, the current study used Tph2-ChR2(C128S) transgenic mice, which express step-type opsin, to allow continuous activation by a pulse of blue light (1.0 s, 473 nm) at the beginning and yellow light (1.0 s, 593 nm) at the end (Fig. 1e). Another difference is that the previous study employed isoflurane, a general gas anesthetic which is also a muscle relaxant, while this experiment employed no anesthetics.

In order to test the possibility that the difference was due to anesthesia, we compared BOLD responses to DRN serotonin stimulation between anesthetized and awake states in the same animals. We employed the same ON/OFF stimulation protocol under anesthesia with 1% isoflurane with the same groups (Transgenic: $n = 7$, WT: $n = 5$, Fig. 3a). We also performed another ON/OFF stimulation session the next day in an awake state in order to eliminate possible confounding by diminished responses after repetitive photo-stimulation (Fig. 3b).

Under anesthesia, we observed mostly negative BOLD responses from the entire cortical region, while BOLD signals in the DRN remained positive (statistical $t > 1$, uncorrected; Fig. 3a & Supplementary S10a). On the other hand, Session 4 in an awake state replicated positive BOLD signals from mid-line cortical regions and DRN (statistical $t > 1$, uncorrected; Fig. 3b & Supplementary Fig. S10b). We then contrasted regional beta values in anesthetized and awake states (Fig. 3c). In the anesthetized state, multiple regions, such as the ACC, dRSC, vRSC, DG, and CA1 showed negative beta values, significantly different from those in an awake state ($p_{FDR-corrected} < 0.05$, correction of multiple comparisons with FDR). We further found a statistically significant difference in peaks of % BOLDs in these regions between anesthetized and awake states ($p_{FDR-corrected} < 0.05$, Fig. 3d; Supplementary S11). Our results support the hypothesis that general anesthesia induces negative BOLD responses by DR serotonin photo-activation.

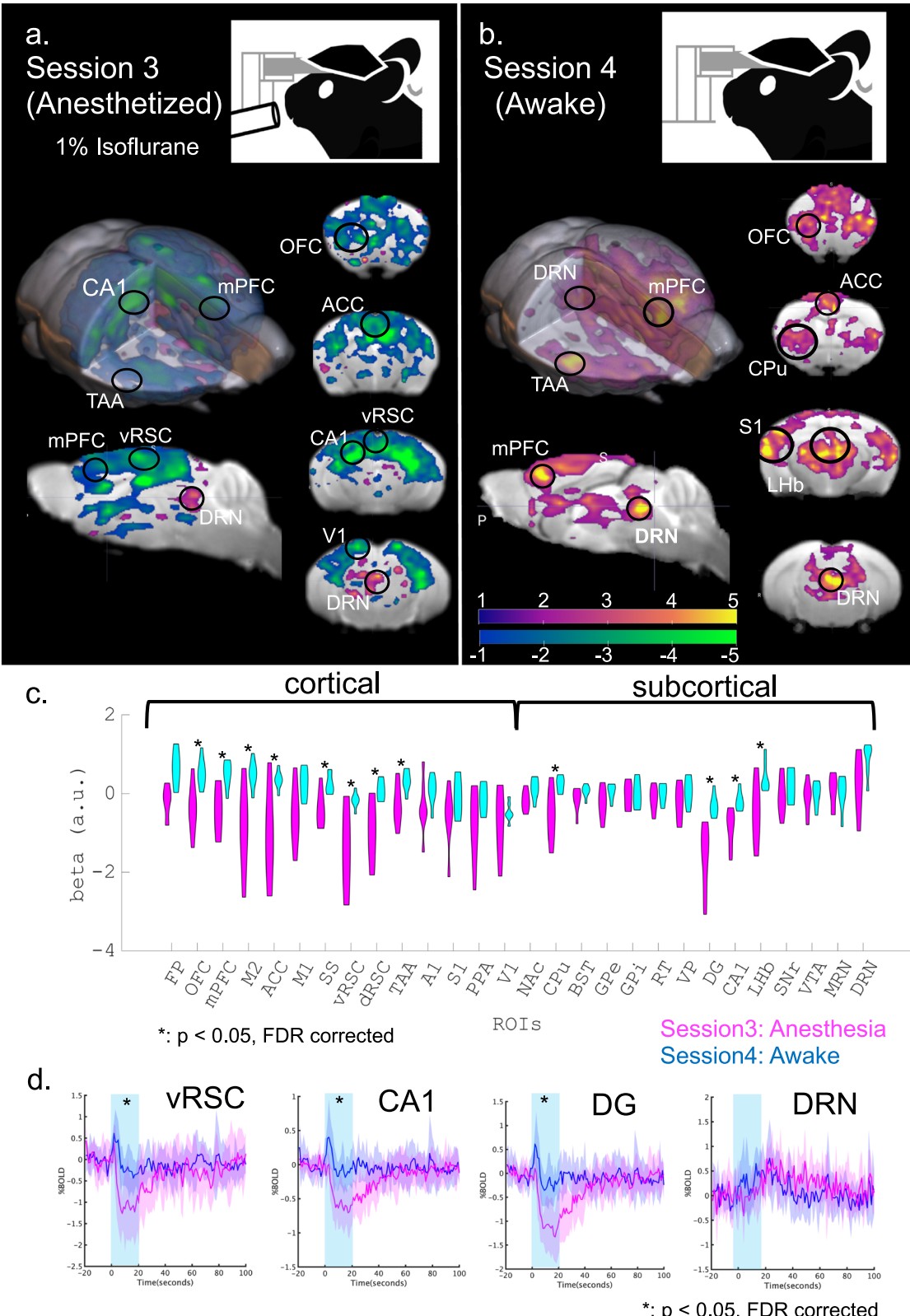

**Brain-wide BOLD responses are explained by serotonin projection density and receptor expression profiles**

Our results revealed a brain-wide profile of evoked BOLD signals by DR serotonin activation (Fig. 2a). One plausible mechanism underlying such a spatial profile is the density of the serotonergic projections from the DRN. In order to test such a possibility, we checked the correlation between the intensity of projections of DRN serotonin

neurons and peak levels of BOLD responses. We extracted serotonin-specific structural projection density from the DRN using the Allen Brain Atlas (Fig. 4a and Supplementary Fig. S12).

Serotonin projection density was statistically correlated with beta values under blue stimulation ($r = 0.462$, $p_{FDR-corrected} < 0.05$), but not under yellow stimulation ($r = 0.132$, $p = 0.529$; Fig. 4b). We also found that projection density was significantly correlated with photo-

**Fig. 3 | Comparison of BOLD responses between anesthetized and awake states.** **a** Optogenetic fMRI session with general anesthesia in Session 3. General anesthesia, 1% isoflurane, was applied to the transgenic group ($n$ = 7) during optogenetic fMRI sessions. GLM analysis following optogenetic stimulation was executed and visualized (T > 1, uncorrected). Magenta and cyan colors indicate positive and negative $t$-values. **b** An optogenetic fMRI session in the awake state in Session 4. As in Session 1, no anesthesia was applied to the transgenic group during optogenetic fMRI sessions. As on day 1, GLM analysis was executed and visualized (T > 1, uncorrected). **c** Comparison between beta values of Sessions 3

and 4 in multiple ROIs. Beta values of blue illumination sessions in Session 3 and Session 4 were statistically contrasted using two-tailed paired Student's $t$-test. * indicates $p_{FDR\text{-corrected}}$ < 0.05 with two-tailed $t$-tests, FDR corrected. **d** Alignment of average % BOLD responses in vRSC, CA1, DG, and DRN in Session 3 from transgenic mice. Blue illumination (1.0 s; 473 nm) was applied at the onset of stimulation, and yellow illumination (1.0 second; 593 nm) followed the offset of stimulation 20 s after the blue illumination. Blue and magenta highlights indicate the range of mean values ± SD. A cyan window indicates stimulation duration for 20 s. * indicates $p_{FDR\text{-corrected}}$ < 0.05 with two-tailed $t$-tests, FDR corrected.

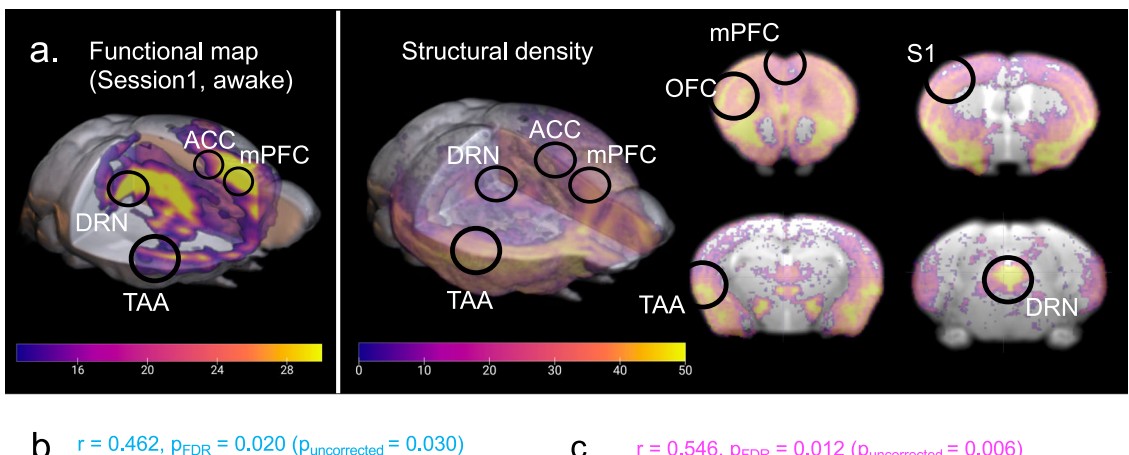

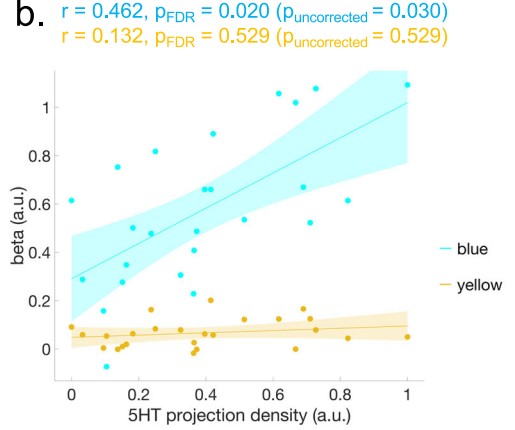

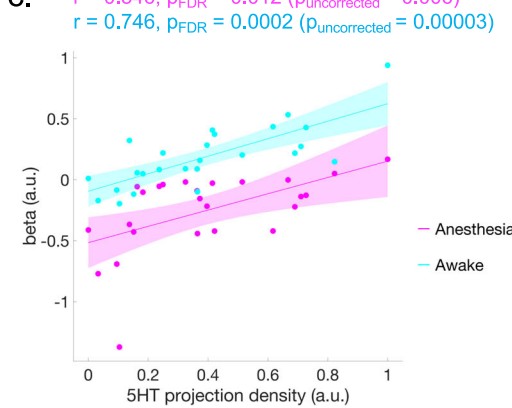

**Fig. 4 | Correlation between the BOLD response and serotonergic projections.** **a** Structural density map of serotonergic projections. A structural density map of serotonergic projections was obtained from the Allen Brain Atlas. The left color bar indicates beta values of BOLD responses corresponding to Fig. 2a. The right color bar indicates normalized intensity of structural density ranging from 0–50. **b** Correlation between projection density and regional beta values. The correlation between regional structural density and average regional beta values across transgenic groups was calculated from Session 1 (blue stimulation: $r$ = 0.462, $p_{FDR}$ = 0.020 ($p_{uncorrected}$ = 0.030); yellow stimulation: $r$ = 0.132, $p_{FDR}$ = 0.529, two-tailed $t$-tests and FDR corrected). Blue and yellow colors indicate data from blue

and yellow illumination sessions, respectively. Cyan and yellow highlights indicate 95% confidence intervals (CI) for correlation coefficients. **c** Correlation between projection density and regional beta values. The correlation between regional structural density and average regional beta values across transgenic groups was calculated (Anesthetized (Session 3): $r$ = 0.546, $p_{FDR\text{-corrected}}$ = 0.012 ($p_{uncorrected}$ = 0.006); Awake (Session 4): $r$ = 0.746, $p_{FDR\text{-corrected}}$ = 0.0002 ($p_{uncorrected}$ = 0.00003), two-tailed $t$-tests and FDR corrected). Magenta and blue colors indicate data from blue illumination sessions in Sessions 3 and 4, respectively. Cyan and magenta highlight indicates 95% CI for correlation coefficients.

stimulated BOLD responses in awake and anesthetized states (Anesthetized (Session 3): $r$ = 0.546, $p_{FDR\text{-corrected}}$ < 0.05; Awake (Session 4): $r$ = 0.758, $p_{FDR\text{-corrected}}$ < 0.001; Fig. 4c).

The previous study revealed that serotonergic structural density was not correlated with the cortical blood volume (CBV) response of DRN serotonin stimulation under 0.5% isoflurane + 0.2 mg/kg/h s.c. medetomidine[15]. We checked whether the difference originated from our ROI difference. After preprocessing with the current study, time series of CBV signals were extracted from 28 ROIs (see "Supplementary Methods"). We first applied GLM to the previous dataset[15]. Although we consistently found negative beta values across multiple brain regions, stimulation of DRN serotonin neurons under anesthesia showed a negative BOLD response,

contrary to the previous result[15] ($p_{FDR\text{-corrected}}$ < 0.05; Supplementary Fig. S13). We then compared the peak intensity of CBV and BOLD signals considering the methodological difference of GLM between these two studies. Pearson correlation between peak intensity of time series from ΔCBV signals and %BOLD responses under anesthesia (Session 3) showed statistical significance ($r$ = 0.53, $p_{uncorrected}$ < 0.01; Supplementary Fig. S14) although neither was significantly correlated with serotonergic structural projection density (peak intensity of CBV signals (Grandjean2019): $r$ = 0.35, $p_{uncorrected}$ = 0.08; peak intensity of BOLD signals (Session 3): $r$ = 0.34, $p_{uncorrected}$ = 0.09; Supplementary Fig. S14).

We further investigated whether photo-stimulated BOLD responses are associated with serotonin synaptic receptors. Gene

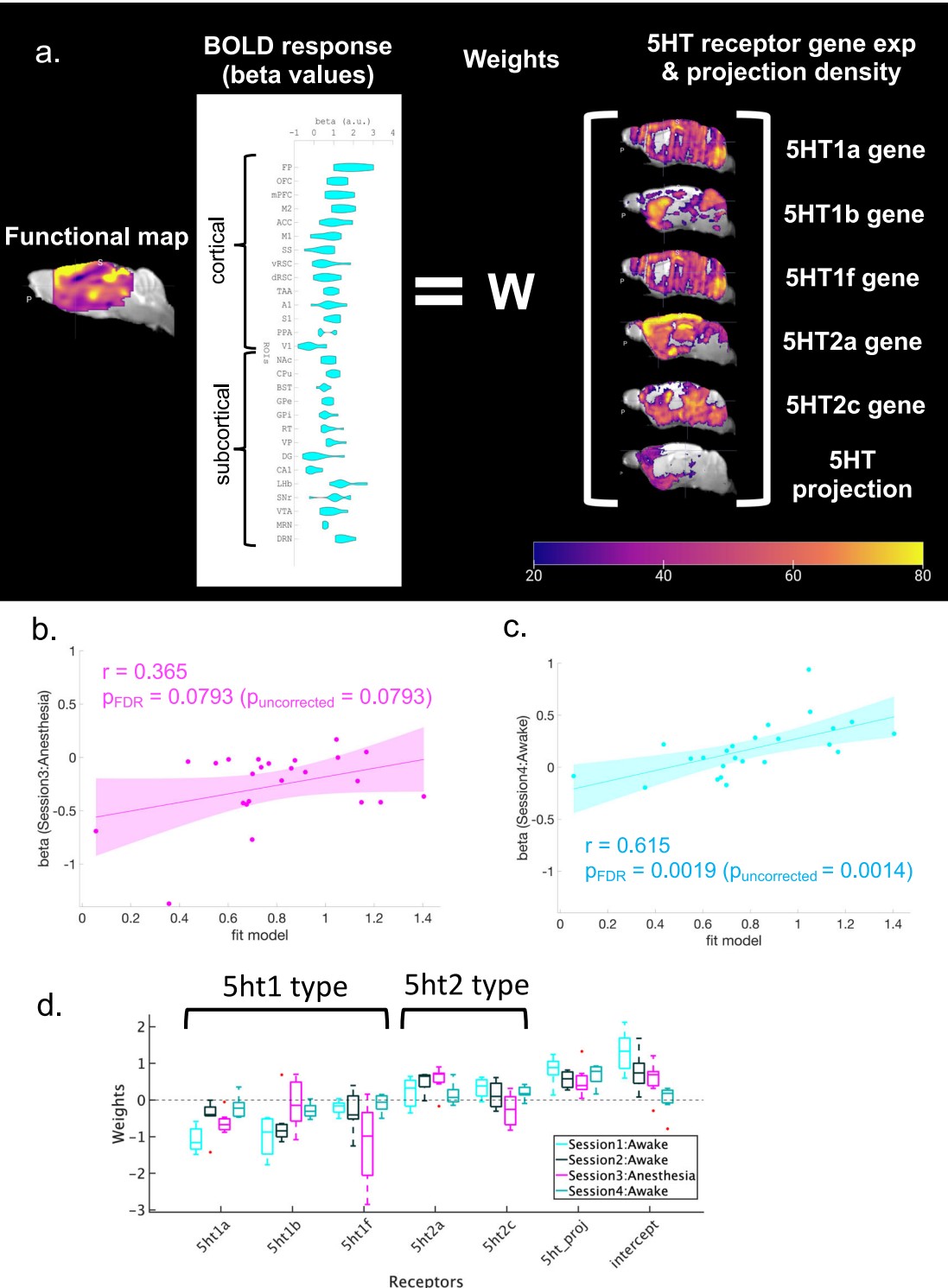

**Fig. 5 | Regression analysis with 5-HT receptors and serotonergic projections.**
**a** Procedure to make a model of predicted beta values in Session 1 of blue illumination. Independent variables, including regional structural densities and regional gene expression of 5ht1a, 5ht1b, 5ht1f, 5ht2a, and 5ht2c were fit with a linear regression model to beta values of multiple ROIs from blue illumination sessions. The right color bar indicates normalized intensity of mRNA expression of 5-HT receptors and the intensity of structural density for projections. The color bar ranges from 0–80. **b** Correlation between modeled beta values from Session 1 and beta values from Session 3 under anesthesia ($r = 0.365$, $p_{\text{FDR-corrected}} = 0.079$, two-tailed $t$-test and FDR corrected). A magenta highlight indicates 95% CI. **c** Correlation

between modeled beta values from Session 1 and beta values from Session 4 (Awake: $r = 0.615$, $p_{\text{FDR-corrected}} = 0.0019$ ($p_{\text{uncorrected}} = 0.0014$), two-tailed $t$-test and FDR corrected). A cyan highlight indicates 95% CI. **d** Weights of fitted models from Session 1, Session 2, and Session 3 under anesthesia, and Session 4 without anesthesia. Each box indicates the range between the 25th percentile (bottom) and the 75th percentile (upper), and a horizontal line within each box is a median value. Whiskers of each box cover 99.3% (±2.7 σ), and each red dot indicates a data point beyond 99.3% coverage. Cyan, black, magenta, and green colors indicate weights of expression profiles from Session 1 ($n = 8$), Session 2 ($n = 7$), Session 3 ($n = 7$) and Session 4 ($n = 7$), respectively.

**Table 2 | Model with parameter estimates of session 1**

| Model statistics (session 1) | | | | |
|---|---|---|---|---|
| | $R^2$ statistic | $F$-statistic | $P$-value (uncorrected) | Estimate of the error variance |
| Session 1 | 0.6603 | 5.5063 | 0.0025 | 0.0643 |

Multiple linear regression was performed to fit parameter estimates from Session 1 (blue stimulation) with the density of serotonergic projection and expression of 5-HT receptors.

expression maps for 5-HT1A, 5-HT1B, 5-HT1F, 5-HT2A, and 5-HT2C receptors were obtained from the Allen Brain Atlas (Supplementary Figs. S12 and S15). We then constructed a multiple linear regression model (MLRM; Fig. 5a) composed of one dependent variable of the BOLD response and six independent variables of the projection density and five gene expression patterns (5-HT1a, 5-HT1b, 5-HT1f, 5-HT2a, and 5-HT2c receptors) after normalization. The model reproduced the BOLD response with $R^2 = 0.66$ ($p_{uncorrected} < 0.01$, Table 2). We also checked the consistency of the model using BOLD responses in different sessions. Using the fit beta values with Session 1, we found a significant correlation with the beta value in Session 4 (Awake: $r = 0.615$, $p_{FDR-corrected} < 0.01$, Fig. 5b), but not in Session 3 (Anesthetized (Session 3): $r = 0.365$, $p_{FDR-corrected} = 0.079$; Fig. 5c).

We investigated the contribution of each receptor by the estimated regression coefficients in anesthetized and awake states (Fig. 5d, Table 3, and Supplementary Table 3). For each subject, we estimated regression coefficients from each session. In the awake state (Sessions 1, 2, and 4), gene expression of 5-HT1 receptors (5-HT1a, 5-HT1b, and 5-HT1f) showed average negative coefficients, indicating negative contributions to BOLD responses. On the other hand, gene expression of 5-HT2 receptors showed average positive coefficients, indicating positive contributions to BOLD responses. We also checked whether the expression of each receptor type differed between awake and anesthetized states using one-way analysis of variance (ANOVA). These results revealed that in the anesthetized state (Session 3), weights for 5-HT1f and 5-HT2c receptors were significantly different ($p_{Bonferroni-corrected} < 0.05$, Table 3).

These results suggest that photo-activated BOLD responses reflect the projection density of serotonin neurons and serotonin receptor expression profiles and that general anesthesia affects responses of 5-HT1 and 5-HT2 receptors differently to induce altered BOLD responses.

## Discussion

Our opto-fMRI investigation of DRN serotonergic modulations of brain-wide dynamics yielded three major discoveries. First, DRN serotonergic activation causes brain-wide positive BOLD responses in an awake state. Second, DRN serotonergic activation under general anesthesia causes brain-wide negative BOLD responses. Third, the whole-brain profile of DRN serotonin-activated BOLD signals is associated with the density of DRN serotonergic projection and expression profiles of 5-HT receptors, with opposite weights for 5-HT1 and 5-HT2 receptors.

The present results, which demonstrate brain-wide activation by DRN serotonergic activation, provide a means of interpreting recordings and manipulation studies in humans and animals[3–5,14,24–28]. Previous studies have demonstrated serotonin's involvement in valence processing[2,26] through modulation of the ACC and ventromedial PFC (vmPFC) on learning signals[27], the CPu on short/long-term prediction of rewards[24], the mPFC and OFC on waiting for delayed rewards[14], and the BST on anxiety-like behaviors[10,21]. A reduction of reward sensitivity is a symptom of major depression. In healthy subjects, 2-week selective SSRI administration potentiated reward signals in the ACC and vmPFC[27]. VTA-projecting DR serotonin neurons induce self-stimulating behaviors, an indication of positive reinforcement[29]. Brain response from brain-wide cortical regions and the VTA may reflect neural

**Table 3 | Results of one-way ANOVA for 5-HT receptor type and structural intensity**

| 5-ht1a | | SS | df | MS | $F$-statistic | $p$-value |
|---|---|---|---|---|---|---|
| | Group | 0.00111 | 1 | 0.00111 | 0 | 0.945 |
| | Error | 6.20697 | 27 | 0.22989 | | |
| | Total | 6.20808 | 28 | | | |
| 5-ht1b | Group | 1.6679 | 1 | 1.66795 | 5.07 | 0.0327 |
| | Error | 8.8794 | 27 | 0.32886 | | |
| | Total | 10.5473 | 28 | | | |
| 5-ht1f | Group | 4.6674 | 1 | 4.66744 | 12.95 | **0.0013*** |
| | Error | 9.7293 | 27 | 0.36034 | | |
| | Total | 14.4967 | 28 | | | |
| 5-ht2a | Group | 0.40786 | 1 | 0.40786 | 3.43 | 0.075 |
| | Error | 3.21128 | 27 | 0.11894 | | |
| | Total | 3.61914 | 28 | | | |
| 5-ht2c | Group | 10.103 | 1 | 10.103 | 31.08 | **0.0019*** |
| | Error | 8.779 | 27 | 0.3251 | | |
| | Total | 18.8809 | 28 | | | |
| Structural density | Group | 0.18596 | 1 | 0.18596 | 1.73 | 0.1999 |
| | Error | 2.90782 | 27 | 0.1077 | | |
| | Total | 3.0938 | 28 | | | |
| Intercept | Group | 0.1135 | 1 | 0.11348 | 0.26 | 0.6171 |
| | Error | 11.9762 | 27 | 0.44356 | | |
| | Total | 12.0896 | 28 | | | |

One-way ANOVA was applied to regression coefficients from each factor to test whether each factor showed statistically different expressions between awake state and anesthetized states. SS, df, and MS indicate the Sum of Squares, Degrees of freedom, and Mean Squares, respectively. * indicates statistical significance after multiple correction with the Bonferroni method ($p_{Bonferroni-corrected} < 0.05$).

mechanisms underlying recovery of reward sensitivity. High-delay discounting of future rewards is another indication of psychiatric disorders, including major depression[30]. Various studies have shown that optogenetic activations of DR serotonergic projections to the orbitofrontal cortex and the mPFC, but not the NAcc, enhance waiting for delayed rewards[3,14]. Our brain response results demonstrate that activations of the OFC and mPFC are induced by optogenetic stimulation of DR serotonin neurons. Anxiety is a general symptom in psychiatric disorders. By increasing extra-synaptic serotonin levels, SSRI contributes to a reduction of anxiety. However, simple activation of DR serotonin neurons did not induce antidepressant effects[11]. Antidepressant roles of DR serotonin neurons on anxiety can be region-dependent[21,31] and work in a time-locked manner[32]. The frontal-cortex- and BST-projecting DR serotonin subsystem promotes antidepressant effects, whereas the amygdala-projecting DR serotonin subsystem enhances anxiogenic effects[21,31]. Nishitani et al. (2018) showed that optogenetic serotonin neurons increase active coping with inescapable stress[32]. Our opto-fMRI results support those studies, as those areas were consistently activated by DRN serotonin stimulation, although such resting-state brain-wide activation also may blunt effects due to co-activations of specific subsystems, without time-locked stimulation.

In addition, average amplitudes of BOLD responses and peak timing are region-dependent. This implies that serotonin-activated BOLD responses are coupled with different regional distributions of receptor densities and serotonin projections, but not with the distance of the activation site. In spite of indirect evidence, our linear approximation of serotonin-activated BOLD responses supports such a possibility. A subpopulation of DR serotonin projections to the cortex is also expressed with the vesicular glutamate transporter Vglut3 (Slc17a8). Co-release of glutamate has recently been reported[6,31]. Our

results may reflect the influence of serotonin as well as glutamate co-release on downstream circuitry.

Serotonergic modulation in short-term windows using optogenetics can avoid limitations of long-term windows, such as pharmacological manipulation[33,34] and chemogenetic manipulation with DREADD (designed receptors exclusively activated by designed drugs)[35]. Multiple pharmacological studies have applied SSRIs to manipulate the serotonin system. Therapeutic benefits of SSRI, including alleviation of depressed mood and psychic anxiety, arise from chronic SSRI administration[36]. Chronic SSRI administration normalizes aberrant activity in the amygdala and ACC during major depressive episodes[37], as well as in brain networks associated with obsessive-compulsive disorder[38]. In healthy individuals, Scholl et al. (2017) found that chronic SSRI administration amplifies both reward and effort signals, fostering learning processes in the ACC, amygdala, mPFC, and striatum[27]. The brain-wide positive response from the ACC, mPFC, and striatum with optogenetic activations in an awake state may partly imitate chronic SSRI administration. On the other hand, acute systemic SSRI administration causes brain-wide inhibitory modulation[33,34]. However, the influence of acute SSRI administration on brain activity can be confounded by the administration period, due to receptor sensitivity[34,39,40]. A previous study stimulated the serotonin system with specifically designed drugs under fMRI[35] and revealed brain-wide positive responses in the somatosensory cortex and subcortical regions, including the hippocampus, the hypothalamus, and the cerebellum. That study did not reveal BOLD responses of mid-line cortical regions, such as the OFC, mPFC, and ACC. A possible explanation for the difference between their results and ours is that BOLD responses cause different time scales, depending on receptor types. Our optogenetic fMRI approach revealed an influence on motivation-related regions at different timings in a transient time window (Supplementary Fig. S5, S6, and S8). Further studies using different methodologies, including optogenetics and chemogenetics, will uncover neural mechanisms underlying different time scales of serotonin modulation.

The mechanism underlying different serotonin modulation between awake and anesthetized states remains unknown. One possible mechanism is a reduction of glutamate transmission via serotonin in target regions[41]. Previous studies have shown that isoflurane suppresses neural activity via inhibition of glutamatergic transmission[41,42]. We found that isoflurane reduced photo-stimulated BOLD signals where they were potentiated in an awake state. Our results also showed that some regions show potentiated negative BOLD responses. In addition to reduced glutamatergic transmission, isoflurane also enhances the sensitivity of GABA neurons[43,44]. Such elevated responses of GABA neurons and reduced responses of glutamatergic neurons may result in diminished positive BOLD signals and potentiated negative BOLD signals. It is interesting to ask whether different anesthetics, like NMDA antagonists, induce different effects on receptor responses while neuron-specific distributions of serotonin receptors are unclear. Serotonin receptors in the prefrontal cortex are differentially distributed[45]. 5-HT1A and 2A receptors are enriched on glutamatergic pyramidal neurons in the PFC[46], whereas 5-HT3A receptors are located on GABAergic interneurons[47]. Since different anesthetics influence different neuron types in various ways, the reactivity of serotonin receptors may induce different responses. In addition, anesthetics may change the effects of serotonin receptors. Some studies showed that isoflurane changes binding affinities of 5-HT$_{2B}$ receptors[48] in vitro and 5-HT$_{1A}$ receptors[49] in vivo. Anesthetics can influence ligand binding to specific subtypes of serotonin receptors.

It is important to consider whether the negative brain-wide response is caused by the neuronal response. Granjean et al. (2019) revealed that DRN serotonin activation reduces the amplitude of local field potentials (LFPs) and burst frequency with multiunit activity (MUA) from a variety of cortical regions including M1, M2, S1, and FA

under 0.5% isoflurane + 0.2 mg/kg/h s.c. medetomidine[15]. That study also examined the association between the brain response to DRN serotonin activation and delta power/burst frequency. This study revealed that a negative brain-wide response is caused by a reduction of cortical neuronal activity under isoflurane & medetomidine. Another study also showed that the negative BOLD response by activation of medial spiny neurons (MSNs) is associated with decreased neuronal activity under isoflurane 0.3–0.7%[50]. There is also the possibility that the negative brain-wide response is due to an increase in cerebral blood flow (CBF), which subsequently reduces neuronal activity under isoflurane. It is well established that isoflurane affects blood vessels and amplifies CBF. Abe et al. (2021) showed that an increase in CBF in the striatum doesn't alter neuronal activity[51]. Furthermore, increasing isoflurane dosages elevates BOLD signal levels[52]. Thus, these studies suggest that the decrease in BOLD signals isn't due to increased CBF. Overall, a brain-wide negative response can be induced by decreased neuronal activity rather than increased CBF.

Furthermore, our results showed a negative BOLD response in the dorsal hippocampus under anesthesia, despite a lack of projections to the hippocampus from the DRN. One potential cause is an indirect influence of the downstream circuitry of DR serotonin neurons. For example, DR serotonin neurons have dense projections to the PFC, which interact with the hippocampus[53,54]. Reduction of the BOLD response in the hippocampus under anesthesia can be indirectly caused by activation of DR serotonergic neurons through their downstream circuitry, although this influence is not observable in awake states. The brain-wide response by activation of DR serotonin neurons under anesthesia may have implications for the sleep state. The similarity of brain states between general anesthesia and non-rapid-eye-movement (NREM) sleep is implied[55]. Previous studies also revealed that DR serotonin neurons mediate the sleep/wake cycle[56,57]. Optogenetic activation of DR serotonin neurons prolonged wakefulness and decreased NREM sleep, whereas their inactivation prevented arousal from $CO_2$-induced sleep[56]. Similarly, optogenetic and chemogenetic activation induced arousal from isoflurane[57]. The negative response under anesthesia may reflect a brain response associated with arousal. Future studies are needed to clarify the neural mechanism underlying BOLD signals under anesthesia.

We also demonstrated that the expression of different 5-HT genes are associated with photo-activated BOLD responses. Discrete contributions of serotonin receptor types influence neural activities, mostly at metabotropic receptors[58]. 5-HT1 receptors are inhibitory metabotropic receptors that suppress neural activity by modulating the cellular signal cascade, whereas 5-HT2 receptors are excitatory metabotropic receptors that enhance neural activity[59,60]. Our MLRM analysis revealed that regression coefficients of gene expression of 5-HT1 and 5-HT2 receptors are negative and positive, respectively. The correspondence between functions of receptor types and signs of coefficients implies that BOLD responses by DRN serotonin photo-activation can be influenced by different receptor distributions. Localization of 5-HT receptors is subtype-dependent. 5-HT$_{1A}$ and 5-HT$_{1B}$ receptors are pre-synaptically expressed, either on somato-dendrites or within axon terminals of DR serotonin neurons[61,62], whereas 5-HT$_{2A}$ and 5-HT$_{2C}$ receptors are predominantly expressed on postsynaptic neurons[63,64]. However, whether 5-HT$_{1F}$ receptors are pre- or post-synaptically expressed is unclear, although they are enriched in cortical layers (layer IV and V) and the caudate putamen[65]. Nonetheless, specifically how 5-HT receptors react to serotonin transmission under general anesthetics is unclear. It is also plausible that general anesthetics change binding affinities of 5-HT receptors[48,49]. As another potential mechanism, serotonin transmission can be influenced by general anesthetics. Some studies revealed that isoflurane lessens spontaneous serotonin levels and spontaneous firing[66,67]. These changes of serotonin transmission and receptor binding affinity may influence responses of 5-HT receptors under anesthesia.

The substantial reduction of magnitude of brain-wide responses between the first, second, and fourth sessions is noteworthy, despite consistent activation patterns. Repeated optogenetic stimulation of DR serotonin neurons and repeated stress potentially account for this reduction. Persistent changes in spontaneous behaviors by repeated optogenetic activation of DR serotonin neurons have been reported[68]. Serotonin induces synaptic plasticity, and repeated stimulation may cause plasticity changes. In addition, repeated restraint stress also may change responses to optogenetic stimulation of DR serotonin neurons, although the habituation protocol to awake fMRI is introduced to prevent restraint stress. Restraint stress increases serotonin release in the amygdala[69]. Molecular and neuronal mechanisms underlying such diminutions are needed to understand the long-term effects of serotonin stimulation.

It is also worth mentioning the limitations of the relationship between the BOLD response and behavioral variability in the present study. Although the present study showed variability of the evoked BOLD response by serotonin stimulation, its behavioral relationship remains unclear. The serotonin system has complementary distinct subsystems[10,14,21,31]. Ren et al. showed that amygdala-projecting DRN serotonin neurons enhance anxiety-like behaviors, whereas frontal-cortex-projecting DRN serotonin neurons enhance anxiolytic behaviors[31]. Miyazaki et al. demonstrated that OFC-projecting serotonin neurons, but not mPFC-projecting serotonin neurons contribute to waiting for delayed rewards[14]. BST-projecting serotonin neurons promote anxiety-related behaviors[10,21,31]. Neural mechanisms underlying diverse ways in which serotonin modulates behavioral functions remain to be clarified.

While our results replicated negative cortical responses under anesthesia and found the association with the CBV response identified in a previous study[15], there are several important differences in methodology and results between previous studies and this one. First, one previous study utilized cerebral blood volume (CBV) signals with a contrast agent to enhance the influence of serotonin modulation, although the current study employed BOLD signals. Although CBV signals are an enhanced proxy of BOLD signals, the methodological difference in imaging may show different responses from various brain regions. Second, we employed a transgenic group targeting the tryptophan hydroxylase 2(Tph2) promoter[11,19] although the previous study employed the ePET-Cre line, targeting transcription factor Pet1[15]. While expression of ChR2 by Tph2 is selective for serotonergic neurons[29], Pet1 promoters are also expressed in non-serotonergic neurons[9,70]. While CA1 and DG were significantly deactivated by serotonin photoactivation in our study, responses in temporal regions were not found in the previous study[15]. The difference in GLM tools between the two studies should be noted. We used SPM for fitting the BOLD response, whereas Grandjean et al. used a different analytic tool, BROCCOLI[71] (https://github.com/wanderine/BROCCOLI), for fitting the CVB response in SPM12. Beta values of the previous dataset using SPM showed negative beta values in the DRN, which did not match ΔCVB signals, although consistently negative brain responses across multiple brain regions were consonant with ΔCVB signals (Supplementary Figs. S13 and S14)[20]. GLM using SPM12 overlooks the negative CBV response. The methodological difference in GLM tools may cause different patterns of functional maps.

Additionally, our results reveal an association between 5HT projection density and BOLD responses from GLM analysis using SPM12, but the previous study did not find this association. One critical methodological difference is that we employed different ROIs from those of the previous study. In the current study, 28 ROIs from the Allen brain atlas included other sources of serotonin and dopamine, the MRN and VTA. In contrast, the previous study used 38 ROIs from the Australian Mouse Brain Mapping Consortium (AMBMC, https://www.imaging.org.au/AMBMC) with multiple ROI hippocampal formations and mid-brain regions. Our analysis revealed an association

between BOLD and CBV responses with our selected 28 ROIs. The ROI difference may yield different results in the association between structural connectivity and responses to optogenetic serotonin activation. Moreover, our results also indicated that neither the peak intensity of BOLD signals nor CBV signals are associated with 5-HT projection density (Supplementary Fig. S14c). Stimulation of the serotonin system and its modulation through metabolic 5-HT receptors promote slow and sustained activities[72,73]. An advantage of GLM analysis is to capture not only acutely activated responses, but also sustained responses by comparing target blocks with other blocks. The correlation of 5-HT projection with peak intensity of brain response may overlook the influence of sustained brain activities. Further studies are required to reveal the association between serotonin projection density in brain regions and the underlying variety of neural responses.

Beside the aforementioned methodological difference, different anesthetics (isoflurane 1% vs. 0.5% isoflurane + 0.2 mg/kg/h s.c. medetomidine), canula locations (horizontal vs. vertical insertion), stimulation power (473 nm laser with 4 mW vs. 4–40 mW), and scanner differences (an 11.7 T MRI scanner with a cryoprobe vs. a 7 T Pharmascan scanner) may also cause distinct brain responses. Further research is required to dissect the differences between the two sets of experimental results.

Testing acute and chronic SSRI treatments on serotonin modulation is important for the clinical modulation of the serotonin system. For example, one study observed an effect of fluoxetine, an SSRI with acutely stressed mice under anesthesia[15]. That study showed enhanced negative responses from brain regions such as the CPu and mPFC. It is unclear whether such effects remain under awake, long-term SSRI administration. It is also important to examine whether reactions of serotonin levels and 5-HT receptors show different dynamics under acute and chronic SSRI administration. A meta-analysis showed that serotonin levels in each region change during the course of SSRI administration[40]. Salvan et al. demonstrated that acute SSRI administration may change the regulation of receptor responses[74]. Further studies on physiological and molecular mechanisms under SSRI administrations are required.

In conclusion, we reported brain-wide responses of DRN serotonin activation via opto-fMRI for the first time in awake animals. We found serotonergic modulation of widespread cortical and subcortical targets at different times as a key controller of brain-wide information transmission. We also showed different modulation by the serotonin system in awake and anesthetized states. We further showed that serotonergic structural projections and gene expression profiles of 5-HT receptors are associated with BOLD responses. Further research is required to understand complex serotonin modulation of brain-wide dynamics, given its neural and molecular mechanisms, such as interactions with neural populations and receptors. Our findings reveal a functional basis for understanding serotonergic control of brain-wide dynamics and functions.

## Methods

All experimental procedures were performed under protocols approved by the Okinawa Institute of Science and Technology Experimental Animal Committee.

### Animals

Serotonin neuron-specific channel rhodopsin 2 variant (C128S) mice were generated by crossing Tph2-tTA mice with tetO-ChR(C128S)-EYFP knock-in mice under control of the tryptophan hydroxylase 2 (Tph2) promoter[11,19]. Eight bi-genic and six wild-type (WT) C57BL/6 J male mice (age > 12 w.o.) were used in this study. The male mice were housed at 24 °C on an inverted dark:light cycle (lights on 10:00-22:00, GMT + 9). After stereotaxic surgery, male mice underwent habituation training[16], MRI experiments, and a reward delay task in turn.

## Stereotaxic surgery

Male mice were initially sedated with 1–3% isoflurane, and were given three types of mixed anesthetic agents ((1 mg/mL) of medetomidine, (5.0 mg/mL) of midazolam, and (5.0 mg/mL) of butorphanol, i.p.)[75]. Cranial skins were removed with a sterilized surgical knife, and a plastic bar ($3 \times 3 \times 27$ mm) for head-fixation in the MRI environment and an optic fiber (480 μm diameter, 4 mm length, Doric Lenses) were fixed to their skulls with dental acrylic cement. The optical fiber was inserted horizontally toward the DRN from the cerebellum in each male mouse. Male Mice were housed separately after surgery and underwent at least a 1-week recovery period. Mice were habituated to the MRI environment for at least another week. They subsequently underwent imaging sessions.

## Acclimation

In order to habituate awake mice to the MRI imaging protocol, an acclimation protocol was employed after the surgeries for head-fixation[16–18]. Male mice were fixed in a mock MRI environment and exposed to MRI imaging sounds and blue illumination for 2 h to habituate them to a leak of illumination during scanning (2.0 Hz). Acclimation to sounds and illumination was repeated 7 days prior to the first imaging session. Following imaging sessions, at least 3 days of acclimation were permitted.

## Functional magnetic resonance imaging (fMRI)

Each animal was placed in the MRI animal bed. T2-weighted structural imaging and three T2* functional imaging sessions were performed. MRI images were acquired with an 11.7-tesla MRI scanner for small animals (Biospec 117/11 system, Bruker Biospin, EmbH, Ettlingen, Germany) with a cryogenic quadrature RF surface probe (Cryoprobe, Bruker BioSpin AG, Fällanden, Switzerland). We performed T2-weighted structural imaging with a RARE sequence with the following parameters: $140 \times 140$ matrix, $13.5 \times 13.5$ mm$^2$ field-of-view, repetition time (TR)/effective echo time (TE) 4000/18.75 (ms), 31 coronal slices, slice thickness: 300 μm.

After T2-weighted structural imaging was performed, a functional MRI session was then executed. A functional MRI session was composed of two imaging runs, a blue-yellow-light stimulation run and a yellow-light stimulation run. All functional imaging was performed using a GE-EPI sequence with the following parameters: $67 \times 67$ matrix, $13.5 \times 13.5$ mm$^2$ field-of-view, TR/TE 1000/10.7 (ms), flip angle: 50°, bandwidth: 333k(Hz), 31 coronal slices and slice thickness: 300 μm, and 690 repetitions.

## ON/OFF stimulation paradigm

A stimulation protocol was composed of ON/OFF stimulation patterns (Fig. 1d). In each activation session, stimulation was initiated 90 volumes after the onset of an imaging session. DRN serotonin neurons were stimulated with blue illumination (490 nm, 3.0 mW, 1.0 s), and stimulation subsequently ceased with yellow illumination (630 nm, 3.0 mW, 1.0 s) 20 s after the onset of blue illumination. This stimulation was repeated 5 times for 120 s after the onset of blue illumination. For the control session, instead of blue illumination, only yellow illumination was applied in stimulation sessions.

## Preprocessing and de-noising

The first 10 volume images of each session were removed from subsequent analysis due to initial imaging instability caused by motion and non-steady-state image quality. 10-fold magnification of images was done to process images with Statistical Parametric Mapping 12 (SPM12), which is designed to process human brain images. Therefore, the voxel size was $2 \times 2 \times 3$ mm in analytical steps. Preprocessing, including motion correction, realignment, co-registration, normalization to a C57BL6/J template[76], and spatial smoothing (kernel with $4 \times 4 \times 6$ mm) was executed with SPM12. Since each slice of a 3D brain

image was taken at a different time, slice-timing correction was applied to correct slice-timing differences by temporal interpolation. Next, realignment of 3D brain images was conducted to correct motion-related changes of brain position. Then, co-registration was executed to overlay functional T2* images onto structural T2 images and to save coordination changes of T2* images for normalization. During normalization, T2 structural images were first warped to fit the average C57BL6/J template[76]. Then T2* functional images were further warped to the template using the co-registered coordination change of the images. In spatial smoothing, voxel signals were spatially smoothed using a Gaussian kernel ($4 \times 4 \times 6$ mm). After pre-processing steps, the general linear model in SPM12 was executed to map spatial activation patterns.

## Regions of interest

The Allen Brain Institute provides parcellation of the C57BL6 brain. In order to extract BOLD signals from anatomically defined brain regions, this parcellation was utilized for subsequent analyses. Parcellation was first warped to the MRI template using Advanced Normalization Tools (ANTs), and 28 regions of interest (ROIs) were selected from the warped parcellation. 28 ROIs were selected with the following three criteria. First, primary cortical regions, especially mid-line regions, were selected. Second, primary reward- and punishment-related sub-cortical regions, including basal ganglia, VTA, and DRN should were included. Third, ventro-lateral cortical and sub-cortical regions, such as the amygdala complex were not selected due to its sensitivity to artifacts near air cavities by the ear canals[77].

## General linear model (GLM) for BOLD signals

A general linear model (GLM) was performed to map beta values of parameter estimation of optogenetic stimulation onto the mouse brain template with SPM12. For statistical functional mapping of optogenetic stimulation, each voxel under an uncorrected $P$-value $< 0.001$ was thresholded, and cluster-wise multiple comparisons were subsequently executed with family-wise error (FWE) by thresholding $p_{FWE} < 0.05$ with probabilistic threshold-free cluster enhancement ($p$TFCE)[78]. Statistical second-level, voxel-wise analysis was finally executed to compare functional maps of blue and yellow illumination from transgenic mice. For Session 1, all procedures were executed accordingly. For Sessions 2, 3, and 4, none of the voxels under uncorrected $P$-values $< 0.001$ survived the thresholding.

## Beta values from BOLD responses

Furthermore, regional beta values were extracted based on ROIs. Statistical significance of regional beta values was subsequently tested with Student's $t$-tests with multiple comparisons with the false discovery rate (FDR; $p$-value (FDR) $< 0.05$).

## Extraction of regional BOLD signal changes

Each regional BOLD signal change (%) is calculated by averaging regional time series on voxels, which are scaled with the mean of time series during the no-stimulation period and followed with multiplication by 100.

## Structural density and RNA expression maps

The structural density map of serotonin connections was obtained from the mouse brain database of the Allen Brain Institute (Experimental IDs: 480074702)[79]. As with the structural density map, gene expression of serotonin receptors was obtained from the mouse brain database of Allen Brain Institute (Experimental IDs: 79556616 (5HT1a), 583 (5HT1b), 69859867 (5HT1f), 81671344 (5HT2a), and 73636098 (5HT2c)). Intensity of the structural density and gene expression was normalized ranging from 0 to 100. Normalized energy intensity was extracted from the normalized atlas. Multiple linear model analysis of

ROIs between betas from GLM analyses and gene expressions was performed.

**Multivariable linear regression model (MLRM)**

Beta values from each session were obtained based on 28 ROIs. Structural density and RNA expression of 5HT1a, 1b, 1f, 2a, and 2c were also extracted based on 28 ROIs. Multivariable linear models were used to estimate weights of structural and gene expression of 5HT receptors; where beta values from each session were the dependent variables, and intensity of structural and gene expression was the explanatory variable. Results of statistical values for fitting are described in Table 2.

**One-way analysis of variance (ANOVA) on receptor contribution**

Regression coefficients of structural density and RNA expression of 5-HT receptors were calculated for each subject for all sessions. One-way ANOVA was applied to regression coefficients of each receptor type in order to test whether receptor expression is statistically different between awake (Sessions 1, 2, & 4) and anesthetized states (Session 3). For analysis, data from all sessions were concatenated (total data: $n = 29$, session 1 (awake): $n = 8$; Session 2(awake): $n = 7$, Session 3 (anesthetized): $n = 7$, Session 4 (awake): $n = 7$). $p$-values from one-way ANOVA were further corrected with the Bonferroni method. Results of ANOVA are described in Supplementary Table ST3.

**A reward delay task**

After imaging sessions, some mice from both groups (Transgenic: $n = 3$, WT: $n = 2$) subsequently performed a waiting experiment for a delayed reward. A sequential tone-food waiting task was utilized as a free operant task[3,4] (Supplementary Fig. S1a). After MRI sessions, individual subjects were trained and tested in an operant-conditioning box (Med-Associates, $12.7 \times 12.7$ cm). In short, subjects were required to initiate trials by nose-poking into a tone site after 500 (ms). The reward-delay task was composed of reward trials (75%) and omission trials (25%). We adopted two types of stimulation protocols (Supplementary Fig. S1b): fixed-delay and random-delay protocols. At the onset of reward and omission trials, either blue or yellow illumination was delivered. In reward trials, blue/yellow-light stimulation was delivered at the onset of each trial until nose-poking was performed, and yellow light was delivered when subjects waited for 6.0 (s) for the fixed protocol and 2, 6, or 10 (s) for the random-delay protocol. However, yellow illumination was also given when subjects failed to wait for the delay. On the other hand, during omission trials, yellow stimulation was given when mice gave up waiting for delayed rewards whereas light stimulation was delivered in the same manner as in the reward trials.

**Reporting summary**

Further information on research design is available in the Nature Portfolio Reporting Summary linked to this article.

## Data availability

Preprocessed data, including functional and structural images, are available from https://openneuro.org (Project_ID: optofMRI_DRN).

## Code availability

In-house codes are also available from https://github.com/Taiyou/OptofMRI_5HT (https://doi.org/10.5281/zenodo.10956557). All software used in this study is also available (Supplementary Table ST1).

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

## Acknowledgements

We thank Dr. Koji Toda for his comments on awake optogenetic stimulation, Dr. Katsuhiko Miyazaki for his comments on the protocol of optogenetic stimulation, Dr. Keigo Hikishima for his comments on fMRI, and Mr. Yoshiaki Ohba for the creation of acclimation tools. We thank Dr. Jiafu Zeng for animal support for MRI acquisition. We also acknowledge Prof. Grandjean for his support in replicating results from a CBV fMRI dataset. This work was supported by Moonshot R&D from the Japan Science and Technology Agency (JST), Japan, grant PMJMS2295 (to H.T.H.). We also acknowledge a grant-in-aid from the Japan Society for the Promotion of Science (JSPS), Japan, grant JP19KK0387 (to N.T.) and JP16H06563 (to K.D.), Transformative Research Areas (A) "Glia decoding" grant 20H05896 (to K.F.T.) and "Unified Theory" grant 23H04975 (to K.D.); Brain Mapping by Integrated Neurotechnologies for Disease Studies (Brain/MINDS) by the Japan Agency for Medical Research and Development (AMED), Japan, grant JP22dm0207069 (to K.F.T.), JP18dm0207030 and 22dm0207001 (to K.D.); a grant from; the Naito Foundation, Japan (to H.T.H.); and research support of Okinawa Institute of Science and Technology Graduate University for the Neural Computation Unit.

## Author contributions

H.T.H., K.F.T. and K.D. conceived the experiments. H.T.H., Y.A. and N.T. designed and prepared experiments. H.T.H. performed MRI experiments. M.T. conducted behavioral experiments. H.T.H. analyzed the results, and H.T.H. wrote the manuscript. H.T.H., Y.A., N.T., M.T., K.F.T. and K.D. reviewed the manuscript.

## Competing interests

The authors declare no competing interests.
