## [Peer Review File · Nature Communications]

Optogenetic activation of dorsal raphe serotonin neurons induces brain-wide activationREVIEWER COMMENTS

Reviewer #1 (Remarks to the Author):

This ms describes effects of optogenetic stimulation of dorsal raphe 5-HT neurons on BOLD activity in awake mice. This work is relevant to understand the role of the DRN 5-HT neurons and also possibly the chronic effects of selective serotonin reuptake inhibitor (SSRI) drugs, of likely clinical relevance. The effects to produce BOLD activation in widespread projection pathways of the DR are contrasted with effects under anaesthesia which are largely opposite in nature and thus generally correspond to those shown by a previous study by other authors in this journal (that hence provided a somewhat misleading picture of the functions of the DR neurons). I was impressed that the present authors carried out this manipulation to simplify comparison of the two studies, which otherwise would have proven potentially confusing for the research community. The authors quantitatively model and relate the degree of BOLD activation to the density of the DR projections and to diverse 5-HT receptor distributions obtained from the Allen database, to provide a generally convincing validating account. Correlation with actual behavioral function would have also been very useful; rather fragmentary data in only a few animals are presented in the Supplementary section to relate the findings to their previous findings on inhibitory control (they describe as 'patience'). Provision of a more complete behavioral dataset in the main text of the ms would have been desirable to strengthen the case that these findings have functional relevance. Overall, the methods and data analysis in this study appear sound and the findings are valuable and of general interest, although it would have been of even greater interest similarly to test effects of acute and chronic SSRI treatment in parallel. I also feel that the title is misleading and simplistic as the loci affected by DR extend to regions other than those simply implicated in 'reward' and punishment and the notion of 'patience' invoked by these authors also implies cognitive control. I had the following specific minor points and queries:

1. The authors need to be more specific about some of their anatomical loci e.g. what do they mean exactly by mPFC (does this include both PL and IL cortex?). OFC (is this lateral OFC and does this region include the insula?).
2. For 5-HT_{1A} receptors, how can the authors parcellate the relative contribution of inhibitory presynaptic and post synaptic 5-HT_{1A} receptors? Can they also more definitively separate the effects mediated via 5HT_{2A} and 5HT_{2C} receptors? Are their actions thus similar under stimulation but opposite with stimulation under anaesthesia?
3. Can the authors clarify in the main text which regions did not exhibit BOLD activation? Do these include the nucleus accumbens, a prominent component of the rodent 'reward system'?
4. Why is there a reduction in in BOLD response hippocampus and related temporal lobe regions under anaesthesia, given that DR neurons do not project there? This should be explained or interpreted.
5. Do the results under anaesthesia have implications for what could be expected in sleep states?
6. Minor. The English is generally excellent, but the final sentence of the abstract should be rephrased.

Reviewer #2 (Remarks to the Author):

Overall: Hamada et al. reported brain-wide fMRI responses to optogenetic stimulation of dorsal raphe serotonin neurons in awake and anesthetized conditions. Performing awake fMRI at an ultrahigh field of 11.7T is impressive. The major new finding is that the BOLD response polarity changed dependent on the brain status. Positive BOLD responses in the stimulation and projection sites were observed at an awake condition, while negative BOLD responses were detected under isoflurane anesthesia. Another important finding is that BOLD responses reflect both serotonin projection density and some subtype receptors. However, enthusiasm is reduced due to relying on correlation, a lack of mechanism studies, and potential experimental design flaw.

Significance of this research: Optogenetic stimulation of the DRN induced fMRI activities in widespread regions (Figure 2) at an awake condition, which is most significant. All active regions

(such as M2, SS) may not involve rewards. Since there are no reward behavioral studies under optogenetic stimulation of specific DRN circuits (e.g., optogenetic stimulation of axon terminals at the presumably reward region), a current manuscript title of "reward network" is misleading.

Similarity and difference with existing literature: Grandjean et al. reported optogenetic fMRI of the DRN under dexmedetomidine/isoflurane anesthesia in Nature Comm. Current manuscript added fMRI responses under an awake condition.

1. Negative BOLD responses were observed under isoflurane anesthesia. This finding is consistent with previous optogenetic studies of the DRN (dexmedetomidine), but in opposite polarity to chemogenetic studies (halothane) (both were cited in this manuscript). Negative BOLD changes can be due to decreased neural activity or suppressed hemodynamic response with increased neural activity. If isoflurane suppresses hemodynamic responses, then increased oxygen metabolism decreases an oxygenation saturation level, resulting in negative BOLD responses. It is crucial to measure neural activities under anesthesia. Please provide underlying mechanisms of negative BOLD responses to the DRN stimulation.

2. Grandjean et al. reported that no correlation between fMRI response and projection density. Here, the projection density is the largest contributor of fMRI responses. How to reconcile these differences (different anesthesia?). Please perform additional experiments with different anesthetics to investigate whether anesthesia is a key issue.

Interpretation:

1. In the correlation analysis between fMRI responses and 5HT projection density, a similar positive correlation was observed in both awake and anesthetized conditions (Fig. 4). Especially, in the anesthetized condition, the area with no projection density has -0.5% BOLD response (intercept). What does this mean biologically?

2. When 6 different variables (receptor expressions and projection density) were used for correlation analysis with beta values of the BOLD response, correlation values (Fig. 5; $r = 0.365$ & 0.615 – this is not same in those in texts) were lower than those of single variable correlation (Fig. 4; $r = 0.546$ & 0.758). Higher correlation values are expected when more variables were used. Please explain why correlation values were reduced.

3. The major difference between awake and anesthetized condition (GABA agonist) was the contribution of 5HT1f and 5HT2c receptors (Fig. 14S). If different anesthetics (e.g., NMDA antagonist) were used, will weighting functions change?

Methodological concerns:

1. Three awake experiments were performed, while one anesthetized condition was used. If anesthesia vs. awake condition was their specific goal, then awake vs. anesthetized condition should be balanced; half of subjects with awake condition first, the other half of subjects with anesthetized condition first. Please comment this issue.

2. In current experimental design, responses (statistical values) of session 1 was higher than those of session 2 (Fig. 3S and Fig. 7S), despite similar peak intensity (Fig. 8S). How and whether can time-dependent reduction of BOLD responses influence current findings? Please comment this.

3. Session 3 and 4 were only one day difference. Are animals in session 4 fully recovered from anesthesia?

4. fMRI activity in the DRN was negligible in session 3 and 4 (Fig. 3D), while positive BOLD response was observed in session 1 and 2 (Fig. 1G). How can this explain?

Minor comments:

1. Fig. 5d, no scale bar for intercept
2. Fig. 5d can be replaced with Fig. 14S
3. Typos: Fig 5s caption: ($n = 8$); Fig. 8c caption

Reviewer #3 (Remarks to the Author):

Hamada et al. reported the mouse whole brain response to optogenetic stimulation of dorsal raphe serotonergic (5-HT) neurons in an awake state. This research investigated how the activation of 5-HT neurons affects their projection areas at the brain-wide level by fMRI. Interestingly, the optogenetic stimulation activated the reward-related areas in awake but inhibited them in the isoflurane-anesthetized state. The main finding manifested that the inhibitory effect of serotonergic neurons, reported by a previous study, should be reconsidered separately from the arousal state. It seems that the differences between awake and anesthetized states are very significant. However, the manuscript has too short explanations, many typos, and odd wording. Moreover, the reviewer has concerns about the inconsistency between the main text and figure legends. Those undermined the reliability. In the present state, this manuscript is inappropriate for publication.

(Lines 92-96) The authors mentioned BOLD responses between transgenic and WT mice in Figure 2a. But "BOLD responses between the blue and yellow illumination" was described in the legend of Figure 2 (lines 508-510). Which is correct?

(Lines 100-102) "We next selected 28 regions of interest (ROIs) from the cortex and subcortical areas that are involved in reward and punishment," If the areas were selected based on the previous studies, the authors should show references.

The reviewer could not find "Table 1" in the manuscript. The abbreviation for ROIs should be described.

(Lines 103-107) Same as Figure 2a. The authors mentioned the comparison between transgenic and WT mice in the main text. But in Figure 2b, the beta values of the blue and yellow illumination in the transgenic mice are graphed. Which is correct?

(Lines 178-181) The details of the multiple linear model (MLM) should be described in the methods.

(Lines 189-197) The authors emphasized the difference in negative and positive weights of each 5-HT receptor. Although the weight values were not the same, the 5-HT1 receptors (5-HT1a, 5-HT1b, and 5-HT1f) showed negative weight, and 5-HT2a showed positive weight constantly. These weight directions were constant in these receptors regardless of awake and anesthetized conditions except for 5-HT2c receptors. The reviewer thinks the results are too weak to mention that "general anesthesia affects responses of 5HT receptors (Line 197)." Since the BOLD responses differ between anesthesia and awake states, the weight values likely changed depending on the conditions.

(Lines 199-202) "that general anesthesia affects responses of 5-HT1 and 5-HT2 type receptors differently to induce altered BOLD responses". The function of the 5-HT receptors might be affected by anesthesia. But presynaptic function(e.g., 5-HT release) may be changed by anesthesia.

(Lines 405-406) The authors should describe the method of GLM in detail.

"parameter stimulation of optogenetic stimulation." What is "stimulation"?

(Line 408) "P value \leq 0.001." What is " \leq "?

(Figures 1g, 2c, 3d, S5, S6, and S11) The reviewer understood that the blue highlighted area in each figure indicates the periods of optogenetic stimulation. In the ON/OFF stimulation paradigm, the blue light was exposed for 1s, and the yellow light was exposed for 20s after the blue light illumination. The blue highlight easily misleads the duration of blue light exposure. The timing and duration of each illumination should be correctly described in these figures like Figure 1e.

(Figures S5 and S8) There are several \ and \$ marks.

Reviewer #4 (Remarks to the Author):

Serotonin (5-HT) is a neuromodulator that is involved in a wide range of cognitive functions. The anatomical organization of the 5-HT system suggests that it acts globally by orchestrating and coordinating the activities of many regions across the entire brain. However, 5-HT research tends to be highly localized, making this work important and timely.

In this manuscript, Hamada et al. measured brain-wide responses to optogenetic stimulation of DRN 5-HT neurons using fMRI in transgenic mice. They found that stimulation caused widespread activation in cortical as well as subcortical regions in awake mice. Interestingly, the same stimulation had an opposite effect in anesthetized mice. Finally, they report correlations between the spatial patterns of DRN stimulation's effects and published spatial expression patterns of different 5-HT receptor types.

The findings of this work are novel and intriguing, particularly the differences between awake and anesthetized conditions. However, I have some concerns regarding this manuscript.

- 1) From figure 2 it seems like DRN stimulation activates pretty much all ROIs. So why does the title of the manuscript single out reward networks when sensory and motor regions are similarly modulated?
- 2) It was very hard for me to understand the statistics. The methods section is confusing and it doesn't specify the details of the tests and when each test is employed. For example, I see multiple-correction, cluster-wise correction and uncorrected tests without further explanation.
- 3) There are many issues with the supplementary section. Some items are not referenced in the main text (for example figure S6, table 2), the naming changes from S2 to 3S, figure 5S doesn't make sense etc.
- 4) There should be a list of brain regions and their corresponding abbreviations (not just the abbreviations).
- 5) It seems like there is a substantial difference between the magnitudes of the responses during the first and second imaging sessions (figure S2). What could account for this change?
- 6) Regarding figure 4 and its interpretation. The possibility of weak excitation under anesthesia could explain lower activations, but not inhibitions relative to baseline. In particular, why do regions that receive little or no 5-HT inputs are the ones that are most inhibited under anesthesia (figure 4c). One possibility is that these are indirect effects that are mediated by other brain regions. This could also explain some discrepancies between this and the study by Grandjean et al. (Ref. 9).
- 7) Regarding the model in figure 5, I couldn't find the statistical methods that were used to test its validity or test whether differences between receptor types were significant.
- 8) It is well established that some DRN 5-HT neurons co-release glutamate. The authors should discuss this fact, since some of the effects of photostimulation may be due to glutamate release.

Minor concerns:

- 1) Figure 4C caption, there is no orange curve.
- 2) I found many typos and grammatical errors. For example:
Line 145: from entire entire
Line 210 density of DRN serotonergic projection density a
Line 331: The optical fiber toward the DRN was horizontally implanted from the cerebellum in each of the mice.
Line 406: stimulation

Reviewer #1 (Comments to the Author):

We thank the reviewer for acknowledging the novelty of our study and for suggestions that improved the manuscript.

This ms describes effects of optogenetic stimulation of dorsal raphe 5-HT neurons on BOLD activity in awake mice. This work is relevant to understand the role of the DRN 5-HT neurons and also possibly the chronic effects of selective serotonin reuptake inhibitor (SSRI) drugs, of likely clinical relevance. The effects to produce BOLD activation in widespread projection pathways of the DR are contrasted with effects under anaesthesia which are largely opposite in nature and thus generally correspond to those shown by a previous study by other authors in this journal (that hence provided a somewhat misleading picture of the functions of the DR neurons). I was impressed that the present authors carried out this manipulation to simplify comparison of the two studies, which otherwise would have proven potentially confusing for the research community. The authors quantitatively model and relate the degree of BOLD activation to the density of the DR projections and to diverse 5-HT receptor distributions obtained from the Allen database, to provide a generally convincing validating account.

Correlation with actual behavioral function would have also been very useful; rather fragmentary data in only a few animals are presented in the Supplementary section to relate the findings to their previous findings on inhibitory control (they describe as 'patience'). Provision of a more complete behavioral dataset in the main text of the ms would have been desirable to strengthen the case that these findings have functional relevance.

It is interesting to observe individual relationship between BOLD response and behavioral variability. Although the present study showed variability of evoked BOLD response by serotonin stimulation, its behavioral relationship remains unclear. Serotonin system is known to have complementary distinct subsystems [13, 30, 38, 49]. Ren et al. showed that amygdala-projecting DRN serotonin neurons enhance anxiety-like behaviors while frontal-cortex-projecting DRN serotonin neurons enhance anxiolytic behaviors [49]. Miyazaki et al., demonstrated that OFC-projecting serotonin neurons but not mPFC-projecting serotonin neurons contribute to waiting for delayed rewards [38]. BST-projecting serotonin neurons promotes anxiety-related behaviors [13, 30, 49]. Neural mechanism underlying different serotonin functions of behaviors also are needs to be clarified. By observing the correlation between BOLD response and behavioral variability, serotonergic modulation on different functions will be clarified. Due to time limitation, we were not able to perform behavioral experiments with all animals. We mentioned this limitation and potential works to examine the relationship between regional response and its function such as anxiety, reward processing and patience in lines 342-352.

Overall, the methods and data analysis in this study appear sound and the findings are valuable and of general interest, although it would have been of even greater interest similarly to test effects of acute and chronic SSRI treatment in parallel.

The study aims to show brain response by DRN serotonergic activation. Testing acute and chronic SSRIs treatment is beyond our original scope. However, we understand that testing acute and chronic SSRIs treatment is also worth examining for clinical interest in serotonin system. For example, Grandjean et al. observed effect of fluoxetine, a SSRI, with acutely stressed mice under anesthesia [9]. They showed enhanced negative responses from brain regions such as CPu and mPFC. It is unclear whether such effects remain under awake and long-term SSRI administration. It is also important to examine whether reactions of serotonin levels and 5-HT receptors show different dynamics by acute and chronic SSRI administrations. A meta-analysis showed that serotonin level in each region changes along the time course of SSRI administration [7]. Salvan et al. demonstrated that acute SSRI administration may change regulations receptor responses [24]. Further studies on physiological and molecular mechanisms under SSRI administrations are required. We mentioned potential works contrasting the effects of acute and chronic SSRI treatments in Lines 379-389.

I also feel that the title is misleading and simplistic as the loci affected by DR extend to regions other than those simply implicated in 'reward' and punishment and the notion of 'patience' invoked by these authors also implies cognitive control.

As the reviewer suggested, evoked response occurred in multiple regions not specific to reward-related regions. Although a previous study supports inhibitory view of DRN serotonin neurons, a core function of the serotonin neurons is reward processing as we mentioned Line 47-48. We would also like to emphasize that optogenetic DRN serotonin activation evoked response from reward-related regions such as CPu, VTA and SNr. However, we understand that specifying evoked brain regions is misleading in the present study. We integrated the reviewer's comment and changed the title, "Optogenetic activation of dorsal raphe serotonin neurons induces brain-wide activation, including reward-related circuits."

I had the following specific minor points and queries:

- 1. The authors need to be more specific about some of their anatomical loci e.g. what do they mean exactly by mPFC (does this include both PL and IL cortex?). OFC (is this lateral OFC and does this region include the insula?).**

We created PFC with infralimbic and prelimbic areas, and OFC includes lateral, medial, ventral, and ventrolateral orbital areas from the Allen Brain Atlas. We summarized the list of ROIs and their abbreviations in Table 1, and detailed correspondence between ROIs and names from the Allen Brain Atlas in Supplementary Table ST2.

- 2. For 5-HT1A receptors, how can the authors parcellate the relative contribution of inhibitory presynaptic and post synaptic 5-HT1A receptors? Can they also more definitively separate the effects mediated via 5HT2A and 5HT2C receptors? Are their actions thus similar under stimulation but opposite with**

stimulation under anaesthesia?

Localization of 5-HT receptors is subtype-dependent. 5-HT_{1A} and 5-HT_{1B} receptors are pre-synaptically expressed either on somatodendrites or within axon terminals of DR serotonin neurons [3, 23]. 5-HT_{2A}, and 5-HT_{2C} receptors are predominantly expressed on postsynaptic neurons [1]. However, whether 5-HT_{1F} receptors are pre- or post-synaptically expressed is unclear, although they are enriched in some brain regions, including cortical layers (layer IV and V) and caudate putamen [14]. Contributions of presynaptic and postsynaptic effects are largely separable by 5-HT receptor types, except for 5-HT_{1F} receptors. As the reviewer pointed out, the mechanism underlying different receptor contributions in awake vs. anesthetized states is also little known. As a potential mechanism, general anesthetics are thought to change binding affinity of 5-HT receptors [16, 29]. Radioligand binding assays showed that 5-HT_{2B} receptors interact with general anesthetics, including isoflurane [16]. In marmoset brains, binding distribution of 5-HT_{1A} receptors showed regional differences under isoflurane [29]. General anesthetics may elicit specific affinity changes on different 5-HT receptors and brain regions. However, further study is needed to determine the mechanism. We mentioned this in the Discussion (Lines 316-328).

- 3. Can the authors clarify in the main text which regions did not exhibit BOLD activation? Do these include the nucleus accumbens, a prominent component of the rodent 'reward system'?**

Our results in the first session revealed no significantly higher response in vRSC, V1, DG, and CA1, whereas other regions, including NAc, showed significantly higher responses. We mentioned this in Lines 115-119.

- 4. Why is there a reduction in in BOLD response hippocampus and related temporal lobe regions under anaesthesia, given that DR neurons do not project there? This should be explained or interpreted.**

Although DR serotonin neurons do not have direct projections to the hippocampus, DR serotonin neurons have brain-wide projections to cortical and subcortical regions. Stimulation of DR serotonin neurons can cause direct and indirect influence on their downstream circuitry. For example, DRN serotonin neurons have dense projections to the PFC [21]. PFC also interacts with the hippocampus [6, 15]. Reduction of response in various brain regions can be caused by DR serotonergic stimulation through indirect influence of such downstream circuitry, although this is not visible in awake states. We mentioned this explanation in the Discussion (Lines 289-295).

- 5. Do the results under anaesthesia have implications for what could be expected in sleep states?**

Although there are known critical differences between sleep and anesthesia, our results under anesthesia may have implications for sleep. The similarity of brain states between general anesthesia and non-rapid eye movement (NREM) sleep is known.

Previous studies revealed that DR serotonin neurons contribute to regulating the sleep/wake cycle [18]. Activation of DR serotonin neurons prolongs awake states, but decrease NREM sleep [10], although their optogenetic inactivation reduces arousal from CO₂-induced sleep [25]. Similarly, activation of DR serotonin neurons promotes recovery from general anesthetics, isoflurane [12]. The negative response under anesthesia may reflect a brain response, associated with arousal. We mentioned this in the Discussion (Lines 297-305).

6. Minor. The English is generally excellent, but the final sentence of the abstract should be rephrased.

We believe that our findings about effects of DR serotonin neurons on brain regions are fundamental and more generally applicable. Only mentioning reward-oriented behaviors may be misleading since multiple brain regions, including reward-related and punishment-related regions are activated. We rephrased our findings in the final sentence (Lines 35-37).

Reviewer #2 (Comments to the Author):

We thank the reviewer for providing detailed comments to enhance the manuscript.

Overall: Hamada et al. reported brain-wide fMRI responses to optogenetic stimulation of dorsal raphe serotonin neurons in awake and anesthetized conditions. Performing awake fMRI at an ultrahigh field of 11.7T is impressive. The major new finding is that the BOLD response polarity changed dependent on the brain status. Positive BOLD responses in the stimulation and projection sites were observed at an awake condition, while negative BOLD responses were detected under isoflurane anesthesia. Another important finding is that BOLD responses reflect both serotonin projection density and some subtype receptors. However, enthusiasm is reduced due to relying on correlation, a lack of mechanism studies, and potential experimental design flaw.

Significance of this research: Optogenetic stimulation of the DRN induced fMRI activities in widespread regions (Figure 2) at an awake condition, which is most significant. All active regions (such as M2, SS) may not involve rewards. Since there are no reward behavioral studies under optogenetic stimulation of specific DRN circuits (e.g., optogenetic stimulation of axon terminals at the presumably reward region), a current manuscript title of “reward network” is misleading.

As the reviewer mentioned, evoked response appeared in multiple regions not specific to reward processing. A critical role of the serotonin neurons is reward processing as we mentioned Line 47-48. We would also like to emphasize that optogenetic DRN serotonin activation evoked response from reward-related regions such as CPu, VTA and SNr. However, we understand the the previous title is misleading from the perspective of specificity. We combined the reviewer's comment and changed the title, “Optogenetic activation of dorsal raphe serotonin neurons induces brain-wide activation, including reward-related circuits.”

Similarity and difference with existing literature: Grandjean et al. reported optogenetic fMRI of the DRN under dexmedetomidine/isoflurane anesthesia in Nature Comm. Current manuscript added fMRI responses under an awake condition.

1. Negative BOLD responses were observed under isoflurane anesthesia. This finding is consistent with previous optogenetic studies of the DRN (dexmedetomidine), but in opposite polarity to chemogenetic studies (halothane) (both were cited in this manuscript). Negative BOLD changes can be due to decreased neural activity or suppressed hemodynamic response with increased neural activity. If isoflurane suppresses hemodynamic responses, then increased oxygen metabolism decreases an oxygenation saturation level, resulting in negative BOLD responses. It is crucial to measure neural activities under anesthesia. Please provide underlying mechanisms of negative BOLD responses to the DRN stimulation.

Using cortical multi-unit recordings under isoflurane, Grandjean et al. revealed that optogenetic DR stimulation reduces spike counts and local field potentials [9]. Thus, we believe that negative BOLD responses under isoflurane represent decreased neural activities. Furthermore, other studies demonstrated that the molecular influence on neural activities by isoflurane was due to inhibition of glutamatergic transmission [1]. Other studies also revealed that sensitivity of GABA receptors increased under isoflurane *in vitro* [4, 28]. We also mentioned these possible mechanisms of negative BOLD responses to the DRN stimulation in the Discussion (Lines 267-277).

2. Grandjean et al. reported that no correlation between fMRI response and projection density. Here, the projection density is the largest contributor of fMRI responses. How to reconcile these differences (different anesthesia?). Please perform additional experiments with different anesthetics to investigate whether anesthesia is a key issue.

We mentioned that one of the main methodological differences between our study and the Grandjean study is the genetic line in the Discussion (Lines 342-364). To confirm this experimentally, we must compare BOLD responses with DRN stimulation in different genetic lines. While such experiments are important, the purpose of our study was to examine brain responses by optogenetic stimulation in awake states. The comparison with the work of [9] was beyond our original interest. Therefore, we believe our study is sufficient to support our conclusions. Since we believe the suggested experiment is important, we discussed potential mechanisms of serotonin influence under anesthesia (Lines 277-287).

Interpretation:

1. In the correlation analysis between fMRI responses and 5HT projection density, a similar positive correlation was observed in both awake and anesthetized conditions (Fig. 4). Especially, in the anesthetized condition, the area with no

projection density has -0.5% BOLD response (intercept). What does this mean biologically?

While dorsal raphe (DR) serotonin neurons do not project directly to the hippocampus, these neurons do have a wide network throughout cortical and subcortical regions. This extensive connectivity allows DR serotonin neurons to exert both direct and indirect impacts on their downstream circuits.

This network includes a dense connection between DR serotonin neurons and the prefrontal cortex (PFC) [21]. Additionally, some studies suggest an interaction between the PFC and the hippocampus [6, 15].

Therefore, it is plausible that stimulation of DR serotonin neurons could result in a reduction of responses in brain regions, such as the hippocampus, through an indirect influence on this downstream circuitry. In our study, these changes were not observed under awake conditions.

For further clarification, we have discussed this potential explanation in detail in the Discussion (Lines 289-295).

2. When 6 different variables (receptor expressions and projection density) were used for correlation analysis with beta values of the BOLD response, correlation values (Fig. 5; $r = 0.365$ & 0.615 – this is not same in those in texts) were lower than those of single variable correlation (Fig. 4; $r = 0.546$ & 0.758). Higher correlation values are expected when more variables were used. Please explain why correlation values were reduced.

The correlation values ($r = 0.365$ (Fig. 5b); $r = 0.615$ (Fig. 5c)) are between fit beta values from session 1 and beta values from session 3/session 4, respectively. Meanwhile, the correlation values, $r = 0.546$ and 0.758 in Fig. 4c, are direct correlation between structural density and beta values from sessions 3 and 4, respectively. In most cases, prediction power (correlation in this case) in a test dataset is lower than prediction power in a training dataset, since regression models fit weights to training datasets.

We clarified these differences in Lines 197-201 and added statistical values for a regression model in Table. 2.

3. The major difference between awake and anesthetized condition (GABA agonist) was the contribution of 5HT1f and 5HT2c receptors (Fig. 14S). If different anesthetics (e.g., NMDA antagonist) were used, will weighting functions change?

Thank you for asking this interesting question. Serotonin receptors, including 5-HT1F and 2C receptors, mediate synaptic activities through GABA and NMDA receptors. Since distributions of serotonin receptors are neuron-dependent, it is possible that different anesthetics can change weighting functions by influencing different types of neurons.

Although neuron-specific distributions of serotonin receptors are still unclear, serotonin receptors in the prefrontal cortex, for example, are differentially distributed [1]. 5-HT_{1A} and 2A receptors are enriched on glutamatergic pyramidal neurons in the PFC [2]. On the other hand, 5-HT_{3A} receptors are located on GABAergic interneurons [20]. Since different anesthetics influence different neuron types, reactivity of serotonin receptors also could be different.

In addition, anesthetics may directly change contributions of serotonin receptors. Some studies demonstrated that binding affinities of 5-HT receptors change under an anesthetized state *in vitro* binding [16] and *in vivo* binding [29]. By direct binding, anesthetics can influence specific subtypes of serotonin receptors. These implications have been mentioned in the Discussion (Lines 277-287).

Methodological concerns:

1. Three awake experiments were performed, while one anesthetized condition was used. If anesthesia vs. awake condition was their specific goal, then awake vs. anesthetized condition should be balanced; half of subjects with awake condition first, the other half of subjects with anesthetized condition first. Please comment this issue.

The main goal of the current study was to study brain responses in awake states. Therefore, the contrast between awake and anesthetized states are beyond our primary interest. However, since our main results under awake states conflicted with the results of Grandjean et al., (2019), we performed the 3rd experiment under anesthesia to check the hypothesis that different brain responses come from genetic difference [9].

The 4th experiment was performed to compensate the possibility that repetitive optogenetic stimulation induces negative response under awake condition. Our results in awake states are consistent across different sessions.

We believe our experiments are sufficient to validate the hypothesis.

2. In current experimental design, responses (statistical values) of session 1 was higher than those of session 2 (Fig. 3S and Fig. 7S), despite similar peak intensity (Fig. 8S). How and whether can time-dependent reduction of BOLD responses influence current findings? Please comment this.

Thank you for this important question. We believe that our findings on brain responses remain valid since brain-wide positive response is consistent across different sessions. Nonetheless, we acknowledge two possible explanations for diminished intensity in brain response across different sessions.

First, repetitive optogenetic stimulation to DRN serotonin neurons may lead to diminished brain response. Correia et al. (2017) demonstrated that daily optogenetic stimulation of DRN serotonin neurons caused increased motor activities in spontaneous

movements [5]. This daily stimulation may cause changes in synaptic plasticity, and different brain response.

Second, acclimation training may induce diminished intensity. Although our method confirmed that 1 week body fixation training reduced stress responses of mice [30], it is plausible that long-term effects of body fixation could impact results [17].

We discussed these potential confounding factors as limitations in the Discussion (Lines 330-340).

3. Session 3 and 4 were only one day difference. Are animals in session 4 fully recovered from anesthesia?

Isoflurane is a gas anesthetic, and mice are fully awake less than 1 h after cessation of isoflurane infusion. It is common to use isoflurane to fixate the mouse body for experiments. Especially, low percentages of isoflurane do not affect mouse wakefulness after 24 h. We believe that one day rest does not influence our results.

4. fMRI activity in the DRN was negligible in session 3 and 4 (Fig. 3D), while positive BOLD response was observed in session 1 and 2 (Fig. 1G). How can this explain?

We appreciate your thoughtful inquiry concerning interpretation of DRN responses, which indeed overlaps with methodological concerns you raised in Question 2.

Our overall interpretation regarding the brain-wide response remains unaffected, as it consistently demonstrates a positive trend across multiple sessions. However, we acknowledge a reduction in the intensity of DRN response.

As outlined in our response to Question 2, factors such as daily optogenetic stimulation or long-term body fixation could potentially contribute to the diminished intensity of DRN responses induced by the optogenetic stimulation of DRN serotonin neurons [5, 17, 19]. A persistent change of spontaneous behaviors by repeated optogenetic activation of DR serotonin neurons is known [5]. Furthermore, repeated body fixation also affects serotonin releases [17]. These factors were potential causes of the distinctive responses between first, second, and fourth sessions.

We've addressed this point in the Discussion (Line 330-340).

Minor comments:

- 1. Fig. 5d, no scale bar for intercept**
- 2. Fig. 5d can be replaced with Fig. 14S**
- 3. Typos: Fig 5s caption: (n = 8); Fig. 8c caption**

We addressed these comments. First, we added bars for intercepts in Fig. 5d. Second, we replaced Fig. 5d with Fig. 14S. Finally, typos in Fig 5S and 8C have been corrected.

Reviewer #3 (Comments to the Author):

We appreciate the reviewer's comments to improve our work.

Hamada et al. reported the mouse whole brain response to optogenetic stimulation of dorsal raphe serotonergic (5-HT) neurons in an awake state. This research investigated how the activation of 5-HT neurons affects their projection areas at the brain-wide level by fMRI. Interestingly, the optogenetic stimulation activated the reward-related areas in awake but inhibited them in the isoflurane-anesthetized state. The main finding manifested that the inhibitory effect of serotonergic neurons, reported by a previous study, should be reconsidered separately from the arousal state. It seems that the differences between awake and anesthetized states are very significant. However, the manuscript has too short explanations, many typos, and odd wording. Moreover, the reviewer has concerns about the inconsistency between the main text and figure legends. Those undermined the reliability. In the present state, this manuscript is inappropriate for publication.

- 1. (Lines 92-96) The authors mentioned BOLD responses between transgenic and WT mice in Figure 2a. But “BOLD responses between the blue and yellow illumination” was described in the legend of Figure 2 (lines 508-510). Which is correct?**

Thank you for pointing out our mistakes. In this study, we compared time series of BOLD signals when DRN is stimulated with blue and yellow light from the transgenic group (n=8), but not between transgenic and WT mice. We revised Lines 97-104 as follows. “A group analysis with a general linear model (GLM) of photo-activated BOLD responses between blue and yellow stimulations from the transgenic group (n = 8)....”

- 2. (Lines 100-102) “We next selected 28 regions of interest (ROIs) from the cortex and subcortical areas that are involved in reward and punishment,” If the areas were selected based on the previous studies, the authors should show references.**

We highlighted our selected standard in Lines 108-111. We selected sources of dopamine and serotonin, including VTA, MRN, and DRN. For a reward system, we selected the primary cortical regions and the basal ganglia, including the caudate putamen, nucleus accumbens, and globus pallidus [1]. For a punishment system, we also selected subcortical regions, including the BST [8], hippocampal complex [11], and the lateral habenula [27]. However, the amygdala was removed from the current study due to a known artifact in the aural cavity [13].

- 3. The reviewer could not find “Table 1” in the manuscript. The abbreviation for ROIs should be described.**

To clarify which brain region is selected, we added a list of ROIs as Table 1 and Supplementary Table ST2.

- 4. (Lines 103-107) Same as Figure 2a. The authors mentioned the comparison between transgenic and WT mice in the main text. But in Figure 2b, the beta values of the blue and yellow illumination in the transgenic mice are graphed. Which is correct?**

As we stated in comment 1 from reviewer 3, we contrasted functional responses between blue and yellow stimuli from transgenic mice (n=8). In Figure 2, beta values between blue and yellow illumination were not compared between transgenic and WT groups.

- 5. (Lines 178-181) The details of the multiple linear model (MLM) should be described in the methods.**

We described the following detail of the multiple linear model (MLM) in the method section.

Multivariable regression models

Beta values from each session were obtained based on 28 ROIs. Structural density and RNA expression from 5HT1a, 1b, 1f, 2a, and 2c were also extracted based on 28 ROIs. Multivariable linear models were used to estimate weights of structural and gene expression of 5HT receptors, in which beta values from each session were the dependent variables, and intensity of structural and gene expressions were the explanatory variables. Results of statistical values for fitting are described in Table 2 (Line 775-777).

- 6. (Lines 189-197) The authors emphasized the difference in negative and positive weights of each 5-HT receptor. Although the weight values were not the same, the 5-HT1 receptors (5-HT1a, 5-HT1b, and 5-HT1f) showed negative weight, and 5-HT2a showed positive weight constantly. These weight directions were constant in these receptors regardless of awake and anesthetized conditions except for 5-HT2c receptors. The reviewer thinks the results are too weak to mention that “general anesthesia affects responses of 5HT receptors (Line 197).” Since the BOLD responses differ between anesthesia and awake states, the weight values likely changed depending on the conditions.**

The result from MLRM revealed that fit model from parameter estimates of session 1 is associated with parameter estimates of session 4, but not the signals from session 3 with anesthetics. Change-of-weight values of anesthetics can be associated with responses of 5-HT receptors.

Nonetheless, as the reviewer mentioned, there are multiple possibilities for weight changes of 5-HT receptors, such as 5-HT release, synaptic response, and receptor affinity by anesthetics. We removed the phrase “general anesthesia affects responses of 5HT receptors (Lines 203-213). We further discussed involvement of detailed possible mechanisms by general anesthetics answering Q7 (Line 316-328).

- 7. (Lines 199-202) “that general anesthesia affects responses of 5-HT1 and 5-HT2**

type receptors differently to induce altered BOLD responses”. The function of the 5-HT receptors might be affected by anesthesia. But presynaptic function(e.g., 5-HT release) may be changed by anesthesia.

As the reviewer mentioned, there are two possible mechanisms of altered responses. First, pre-/post- synaptic functions of 5-HT1 and 2 receptors can be influenced by isoflurane. Localization of 5-HT receptors is subtype-dependent. 5-HT2A and 5-HT2C receptors are predominantly expressed on postsynaptic neurons [1]. Meanwhile, whether 5-HT1F receptors are pre- or post-synaptically expressed is unclear, although they are enriched in cortical layers (layer IV and V) and caudate putamen [14] Since isoflurane is a GABA agonist, responses of these receptors can be affected through GABAergic interneurons. Other studies also revealed that general anesthetics change binding affinity of 5-HT receptors [16, 29]. Influences of these different localizations through GABAergic neurons and binding affinity can affect functions of 5-HT receptors.

Second, serotonin transmission in the DRN can be affected by isoflurane. It is plausible that isoflurane influences 5-HT release through GABAergic interneurons within the DRN. Isoflurane reduces spontaneous serotonin levels and spontaneous firing [1].

We discussed these influences of synaptic functions and 5-HT release in the Discussion (Line 316-328).

- 8. (Lines 405-406) The authors should describe the method of GLM in detail. “parameter estimation of optogenetic stimulation.” What is “stimation”?**

There was a typo in this sentence and we revised it as “estimation”. We meant to describe parameter estimation for beta values in GLM (Line 496).

- 9. (Line 408) “P value ; 0.001.” What is “;”?**

We revised it as “<“ in Line 498.

- 10. (Figures 1g, 2c, 3d, S5, S6, and S11) The reviewer understood that the blue highlighted area in each figure indicates the periods of optogenetic stimulation. In the ON/OFF stimulation paradigm, the blue light was exposed for 1s, and the yellow light was exposed for 20s after the blue light illumination. The blue highlight easily misleads the duration of blue light exposure. The timing and duration of each illumination should be correctly described in these figures like Figure 1e.**

We clarified the difference between duration of blue illumination (1 s) and highlighted stimulation period (20 s). For each caption, we mentioned "Blue illumination (1.0 s; 473 nm) was applied at the onset of stimulation, and yellow illumination (1.0 second; 593 nm) followed the offset of stimulation 20 s after blue illumination. A cyan window indicates stimulation duration for 20 seconds."

Reviewer #4 (Comments to the Author):

We thank the reviewer for suggestions to improve our manuscript.

Serotonin (5-HT) is a neuromodulator that is involved in a wide range of cognitive functions. The anatomical organization of the 5-HT system suggests that it acts globally by orchestrating and coordinating the activities of many regions across the entire brain. However, 5-HT research tends to be highly localized, making this work important and timely.

In this manuscript, Hamada et al. measured brain-wide responses to optogenetic stimulation of DRN 5-HT neurons using fMRI in transgenic mice. They found that stimulation caused widespread activation in cortical as well as subcortical regions in awake mice. Interestingly, the same stimulation had an opposite effect in anesthetized mice. Finally, they report correlations between the spatial patterns of DRN stimulation's effects and published spatial expression patterns of different 5-HT receptor types.

The findings of this work are novel and intriguing, particularly the differences between awake and anesthetized conditions. However, I have some concerns regarding this manuscript.

- 1. From figure 2 it seems like DRN stimulation activates pretty much all ROIs. So why does the title of the manuscript single out reward networks when sensory and motor regions are similarly modulated?**

Thank you for pointing out that brain responses are induced not only by reward networks, but also other brain regions. We revised the title to, "Optogenetic activation of dorsal raphe serotonin neurons induces brain-wide activation, including reward-related circuits."

- 2. It was very hard for me to understand the statistics. The methods section is confusing and it doesn't specify the details of the tests and when each test is employed. For example, I see multiple-correction, cluster-wise correction and uncorrected tests without further explanation.**

For the imaging dataset, we clarified steps of statistical functional maps of BOLD responses. First, functional maps were thresholded with uncorrected P value < 0.001 . Second, cluster-wise correction was executed with family-wise error (FWE) by thresholding $p_{FWE} < 0.05$ with probabilistic threshold-free cluster enhancement (pTFCE) [26].

For session 1, all steps were executed accordingly. However, for sessions 2-4, none of the voxels with uncorrected P values < 0.001 remained after thresholding. We contrasted beta values with Student's t test with multiple comparison with the false discovery rate (FDR; p value (FDR) < 0.05).

We added the statistical methods in the main text (Lines 494-504). We further mentioned the General Linear Model (GLM) for BOLD signals and Beta values from BOLD responses.

- 3. There are many issues with the supplementary section. Some items are not referenced in the main text (for example figure S6, table 2), the naming changes from S2 to 3S, figure 5S doesn't make sense etc.**

We revised the labels of Supplementary Figures, now denoting them as Supplementary Figures S1 through S13. We also confirmed that all figures are referenced in the main text.

- 4. There should be a list of brain regions and their corresponding abbreviations (not just the abbreviations).**

We added a list of brain regions and their abbreviations as Table 1.

- 5. It seems like there is a substantial difference between the magnitudes of the responses during the first and second imaging sessions (figure S2). What could account for this change?**

Thank you for pointing out this difference. We can also see a substantial reduction of responses between the second and fourth imaging sessions. There are several potential causes of the substantial effect. One potential cause of decreased responses is repeated photo-stimulation of DRN serotonin neurons. A previous study demonstrated that transient and repeated photo-stimulation of DRN serotonin neurons induced differential effects on spatial locomotion [5]. Another cause of responses is stress through repeated fMRI sessions in awake conditions. Repeated body restrictions during fMRI sessions may create chronic stress. Previous studies revealed that chronic stress influences responses of serotonin neurons through 5-HT receptors [17, 19]. We have mentioned these issues in the Discussion (Line 330-341).

- 6. Regarding figure 4 and its interpretation. The possibility of weak excitation under anesthesia could explain lower activations, but not inhibitions relative to baseline. In particular, why do regions that receive little or no 5-HT inputs are the ones that are most inhibited under anesthesia (figure 4c). One possibility is that these are indirect effects that are mediated by other brain regions. This could also explain some discrepancies between this and the study by Grandjean et al. (Ref. 9).**

Thank you for suggesting this helpful interpretation. One potential explanation, as you suggested, is that general anesthetics change responses to DR serotonin inputs, and can cause indirect responses mediated by other brain regions. For example, DR serotonin neurons project to the PFC [21-22], which also interacts with the hippocampus [6, 15]. Reduction of BOLD responses in the hippocampus under anesthesia can indirectly influence the hippocampus via downstream circuitry of DR serotonin neurons. We discussed this in Lines 289-295.

7. Regarding the model in figure 5, I couldn't find the statistical methods that were used to test its validity or test whether differences between receptor types were significant.

For Figures 5b-c, we first fit structural density and RNA expression of 5-HT receptors to parameter estimates from session 1 with a multiple linear regression model. For Figure 5b, we calculated the Pearson's correlation coefficient between the fit model and parameter estimates from session 4 (Awake), whereas the Pearson's correlation coefficient between the fit model and parameter estimates from session 3 (Anesthetized) was calculated for Figure 5c. We clarified the regression model for the Figure 5b-c (Line 499-505) and model statistics from a multiple linear regression model with session 1 (Table 2).

For Figure 5d, we previously demonstrated a mere qualitative analysis on receptor contribution. Therefore, we conducted analysis of variance (ANOVA) to test contributions of each receptor type. We first calculated regression coefficients from each subject from sessions 1-4 and demonstrated that one-way variance of analysis (ANOVA) for receptor to check whether receptor contribution is different between awake and anesthetized states. These results revealed statistically significant differences in expression of 5-HT1f and 5-HT2c receptors between awake and anesthetized states ($p < 0.05$, Bonferroni corrected). We clarified the method in the section "One-way Analysis of Variance (ANOVA) on Receptor Contribution", and modified main sentences (Lines 507-515). Furthermore, we replaced Figure 5d and added statistical descriptions for each ANOVA in Table 3.

Figure 5d. Weights of the fit models from session 1, session 2, and session 3 under anesthesia, and session 4 without anesthesia. Cyan, black, magenta, and green colors indicate weights of expression profiles from session 1 (n=8), session2 (n=7), session 3 (n=7) and session 4 (n=7), respectively.

8. It is well established that some DRN 5-HT neurons co-release glutamate. The authors should discuss this fact, since some of the effects of photostimulation may be due to glutamate release.

As the reviewer commented, a subpopulation of DR serotonin projections to the

cortex, which are co-expressed with vesicular glutamate transporter 3 (VGluT3), glutamate has been reported (Ren et al., 2018; Ren et al., 2019). Findings of brain-wide responses, especially cortical regions, may reflect influence by serotonin and glutamate transmission. We discussed this in the Discussion (Lines 241-244).

Minor concerns:

1. Figure 4C caption, there is no orange curve.

We removed ‘orange’ and added ‘magenta’.

2. I found many typos and grammatical errors. For example:

Line 145: from entire entire

Line 210 density of DRN serotonergic projection density a

Line 331: The optical fiber toward the DRN was horizontally implanted from the cerebellum in each of the mice.

Line 406: stimation

We revised these passages as suggested.

Line 159: from the entire cortical region

Line 226: density of DRN serotonergic projections and

Line 427: The optical fiber was inserted horizontally toward the DRN from the cerebellum in each mouse.

Line 497: estimation

References

- Amargos-Bosch, M., Bortolozzi, A., Puig, M. V., Serrats, J., Adell, A., Celada, P., Toth, M., Mengod, G., & Artigas, F. (2004). Co-expression and in vivo interaction of serotonin1A and serotonin2A receptors in pyramidal neurons of prefrontal cortex. *Cereb Cortex*, 14(3), 281-299. <https://doi.org/10.1093/cercor/bhg128>
- Boschert, U., Amara, D. A., Segu, L., & Hen, R. (1994). The mouse 5-hydroxytryptamine1B receptor is localized predominantly on axon terminals. *Neuroscience*, 58(1), 167-182. [https://doi.org/10.1016/0306-4522\(94\)90164-3](https://doi.org/10.1016/0306-4522(94)90164-3)
- Caraiscos, V. B., Newell, J. G., You-Ten, K. E., Elliott, E. M., Rosahl, T. W., Wafford, K. A., MacDonald, J. F., & Orser, B. A. (2004). Selective enhancement of tonic GABAergic inhibition in murine hippocampal neurons by low concentrations of the volatile anesthetic isoflurane. *J Neurosci*, 24(39), 8454-8458. <https://doi.org/10.1523/JNEUROSCI.2063-04.2004>
- Cornea-Hebert, V., Riad, M., Wu, C., Singh, S. K., & Descarries, L. (1999). Cellular and subcellular distribution of the serotonin 5-HT2A receptor in the central nervous system of adult rat. *J Comp Neurol*, 409(2), 187-209. [https://doi.org/10.1002/\(sici\)1096-9861\(19990628\)409:2<187::aid-cne2>3.0.co;2-p](https://doi.org/10.1002/(sici)1096-9861(19990628)409:2<187::aid-cne2>3.0.co;2-p)
- Correia, P. A., Lottem, E., Banerjee, D., Machado, A. S., Carey, M. R., & Mainen, Z. F. (2017). Transient inhibition and long-term facilitation of locomotion by phasic optogenetic activation of serotonin neurons. *Elife*, 6. <https://doi.org/10.7554/eLife.20975>

- Eichenbaum, H. (2017). Prefrontal-hippocampal interactions in episodic memory. *Nat Rev Neurosci*, 18(9), 547-558. <https://doi.org/10.1038/nrn.2017.74>
- Garcia-Garcia, A. L., Canetta, S., Stujenske, J. M., Burghardt, N. S., Ansorge, M. S., Dranovsky, A., & Leonardo, E. D. (2018). Serotonin inputs to the dorsal BNST modulate anxiety in a 5-HT(1A) receptor-dependent manner. *Mol Psychiatry*, 23(10), 1990-1997. <https://doi.org/10.1038/mp.2017.165>
- Giorgetti, M., & Tecott, L. H. (2004). Contributions of 5-HT(2C) receptors to multiple actions of central serotonin systems. *Eur J Pharmacol*, 488(1-3), 1-9. <https://doi.org/10.1016/j.ejphar.2004.01.036>
- Grandjean, J., Corcoba, A., Kahn, M. C., Upton, A. L., Deneris, E. S., Seifritz, E., Helmchen, F., Mann, E. O., Rudin, M., & Saab, B. J. (2019). A brain-wide functional map of the serotonergic responses to acute stress and fluoxetine. *Nat Commun*, 10(1), 350. <https://doi.org/10.1038/s41467-018-08256-w>
- Ito, H., Yanase, M., Yamashita, A., Kitabatake, C., Hamada, A., Suhara, Y., Narita, M., Ikegami, D., Sakai, H., Yamazaki, M., & Narita, M. (2013). Analysis of sleep disorders under pain using an optogenetic tool: possible involvement of the activation of dorsal raphe nucleus-serotonergic neurons. *Mol Brain*, 6, 59. <https://doi.org/10.1186/1756-6606-6-59>
- Izquierdo, I., Furini, C. R., & Myskiw, J. C. (2016). Fear Memory. *Physiol Rev*, 96(2), 695-750. <https://doi.org/10.1152/physrev.00018.2015>
- Li, A., Li, R., Ouyang, P., Li, H., Wang, S., Zhang, X., Wang, D., Ran, M., Zhao, G., Yang, Q., Zhu, Z., Dong, H., & Zhang, H. (2021). Dorsal raphe serotonergic neurons promote arousal from isoflurane anesthesia. *CNS Neurosci Ther*, 27(8), 941-950. <https://doi.org/10.1111/cns.13656>
- Li, R., Liu, X., Sidabras, J. W., Paulson, E. S., Jesmanowicz, A., Nencka, A. S., Hudetz, A. G., & Hyde, J. S. (2015). Restoring susceptibility induced MRI signal loss in rat brain at 9.4 T: A step towards whole brain functional connectivity imaging. *PLOS ONE*, 10(4), e0119450. <https://doi.org/10.1371/journal.pone.0119450>
- Lucaites, V. L., Krushinski, J. H., Schaus, J. M., Audia, J. E., & Nelson, D. L. (2005). [3H]LY334370, a novel radioligand for the 5-HT1F receptor. II. Autoradiographic localization in rat, guinea pig, monkey and human brain. *Naunyn Schmiedebergs Arch Pharmacol*, 371(3), 178-184. <https://doi.org/10.1007/s00210-005-1036-8>
- Malik, R., Li, Y., Schamiloglu, S., & Sohal, V. S. (2022). Top-down control of hippocampal signal-to-noise by prefrontal long-range inhibition. *Cell*, 185(9), 1602-1617 e1617. <https://doi.org/10.1016/j.cell.2022.04.001>
- Massey, C. A., Iceman, K. E., Johansen, S. L., Wu, Y., Harris, M. B., & Richerson, G. B. (2015). Isoflurane abolishes spontaneous firing of serotonin neurons and masks their pH/CO(2) chemosensitivity. *J Neurophysiol*, 113(7), 2879-2888. <https://doi.org/10.1152/jn.01073.2014>
- Matsunaga, F., Gao, L., Huang, X. P., Saven, J. G., Roth, B. L., & Liu, R. (2015). Molecular interactions between general anesthetics and the 5HT2B receptor. *J Biomol Struct Dyn*, 33(1), 211-218. <https://doi.org/10.1080/07391102.2013.869483>
- Mo, B., Feng, N., Renner, K., & Forster, G. (2008). Restraint stress increases serotonin release in the central nucleus of the amygdala via activation of corticotropin-

- releasing factor receptors. *Brain Res Bull*, 76(5), 493-498.
<https://doi.org/10.1016/j.brainresbull.2008.02.011>
- Oikonomou, G., Altermatt, M., Zhang, R. W., Coughlin, G. M., Montz, C., Gradinaru, V., & Prober, D. A. (2019). The Serotonergic Raphe Promote Sleep in Zebrafish and Mice. *Neuron*, 103(4), 686-701 e688.
<https://doi.org/10.1016/j.neuron.2019.05.038>
- Prakash, N., Stark, C. J., Keisler, M. N., Luo, L., Der-Avakian, A., & Dulcis, D. (2020). Serotonergic Plasticity in the Dorsal Raphe Nucleus Characterizes Susceptibility and Resilience to Anhedonia. *J Neurosci*, 40(3), 569-584.
<https://doi.org/10.1523/JNEUROSCI.1802-19.2019>
- Puig, M. V., Santana, N., Celada, P., Mengod, G., & Artigas, F. (2004). In vivo excitation of GABA interneurons in the medial prefrontal cortex through 5-HT₃ receptors. *Cereb Cortex*, 14(12), 1365-1375.
<https://doi.org/10.1093/cercor/bhh097>
- Puig, M. V., & Gullledge, A. T. (2011). Serotonin and prefrontal cortex function: neurons, networks, and circuits. *Mol Neurobiol*, 44(3), 449-464.
<https://doi.org/10.1007/s12035-011-8214-0>
- Ren, J., Friedmann, D., Xiong, J., Liu, C. D., Ferguson, B. R., Weerakkody, T., DeLoach, K. E., Ran, C., Pun, A., Sun, Y., Weissbourd, B., Neve, R. L., Huguenard, J., Horowitz, M. A., & Luo, L. (2018). Anatomically Defined and Functionally Distinct Dorsal Raphe Serotonin Sub-systems. *Cell*, 175(2), 472-487 e420. <https://doi.org/10.1016/j.cell.2018.07.043>
- Ren, J., Isakova, A., Friedmann, D., Zeng, J., Grutzner, S. M., Pun, A., Zhao, G. Q., Kolluru, S. S., Wang, R., Lin, R., Li, P., Li, A., Raymond, J. L., Luo, Q., Luo, M., Quake, S. R., & Luo, L. (2019). Single-cell transcriptomes and whole-brain projections of serotonin neurons in the mouse dorsal and median raphe nuclei. *Elife*, 8. <https://doi.org/10.7554/eLife.49424>
- Riad, M., Garcia, S., Watkins, K. C., Jodoin, N., Doucet, E., Langlois, X., el Mestikawy, S., Hamon, M., & Descarries, L. (2000). Somatodendritic localization of 5-HT_{1A} and preterminal axonal localization of 5-HT_{1B} serotonin receptors in adult rat brain. *J Comp Neurol*, 417(2), 181-194.
<https://www.ncbi.nlm.nih.gov/pubmed/10660896>
- Sesack, S. R., & Grace, A. A. (2010). Cortico-Basal Ganglia reward network: microcircuitry. *Neuropsychopharmacology*, 35(1), 27-47.
<https://doi.org/10.1038/npp.2009.93>
- Smith, H. R., Leibold, N. K., Rappoport, D. A., Ginapp, C. M., Purnell, B. S., Bode, N. M., Alberico, S. L., Kim, Y. C., Audero, E., Gross, C. T., & Buchanan, G. F. (2018). Dorsal Raphe Serotonin Neurons Mediate CO₂-Induced Arousal from Sleep. *J Neurosci*, 38(8), 1915-1925. <https://doi.org/10.1523/JNEUROSCI.2182-17.2018>
- Spisak, T., Spisak, Z., Zunhammer, M., Bingel, U., Smith, S., Nichols, T., & Kincses, T. (2019). Probabilistic TFCE: A generalized combination of cluster size and voxel intensity to increase statistical power. *Neuroimage*, 185, 12-26.
<https://doi.org/10.1016/j.neuroimage.2018.09.078>
- Trusel, M., Nuno-Perez, A., Lecca, S., Harada, H., Lalive, A. L., Congiu, M., Takemoto, K., Takahashi, T., Ferraguti, F., & Mameli, M. (2019). Punishment-Predictive Cues Guide Avoidance through Potentiation of Hypothalamus-to-

- Habenula Synapses. *Neuron*, 102(1), 120-127 e124.
<https://doi.org/10.1016/j.neuron.2019.01.025>
- Wang, H. Y., Eguchi, K., Yamashita, T., & Takahashi, T. (2020). Frequency-Dependent Block of Excitatory Neurotransmission by Isoflurane via Dual Presynaptic Mechanisms. *J Neurosci*, 40(21), 4103-4115.
<https://doi.org/10.1523/JNEUROSCI.2946-19.2020>
- Whittington, R. A., & Virag, L. (2006). Isoflurane decreases extracellular serotonin in the mouse hippocampus. *Anesth Analg*, 103(1), 92-98, table of contents.
<https://doi.org/10.1213/01.ane.0000221488.48352.61>
- Winegar, B. D., & MacIver, M. B. (2006). Isoflurane depresses hippocampal CA1 glutamate nerve terminals without inhibiting fiber volleys. *BMC Neurosci*, 7, 5.
<https://doi.org/10.1186/1471-2202-7-5>
- Ying, S. W., Werner, D. F., Homanics, G. E., Harrison, N. L., & Goldstein, P. A. (2009). Isoflurane modulates excitability in the mouse thalamus via GABA-dependent and GABA-independent mechanisms. *Neuropharmacology*, 56(2), 438-447. <https://doi.org/10.1016/j.neuropharm.2008.09.015>
- Yokoyama, C., Mawatari, A., Kawasaki, A., Takeda, C., Onoe, K., Doi, H., Newman-Tancredi, A., Zimmer, L., & Onoe, H. (2016). Marmoset Serotonin 5-HT1A Receptor Mapping with a Biased Agonist PET Probe 18F-F13714: Comparison with an Antagonist Tracer 18F-MPPF in Awake and Anesthetized States. *Int J Neuropsychopharmacol*, 19(12). <https://doi.org/10.1093/ijnp/pyw079>
- Yoshida, K., Mimura, Y., Ishihara, R., Nishida, H., Komaki, Y., Minakuchi, T., Tsurugizawa, T., Mimura, M., Okano, H., Tanaka, K. F., & Takata, N. (2016). Physiological effects of a habituation procedure for functional MRI in awake mice using a cryogenic radiofrequency probe. *J Neurosci Methods*, 274, 38-48.
<https://doi.org/10.1016/j.jneumeth.2016.09.013>

REVIEWER COMMENTS

Reviewer #1 (Remarks to the Author):

The revision has diligently addressed my comments, though perhaps not always in the most satisfactory ways.

1. I really think the emphasis on the reward system, for example even in the title of the ms, is inappropriate. The selection of caudate-putamen as part of the reward system is arbitrary, as Delgado amongst others clearly shows that the dorsal striatum can also be part of the 'punishment system'. It is perhaps also naive to describe the pallidum simply as part of the reward system. The hippocampus is implicated in appetitive functions as well as punishment. The frontal cortex clearly has other functions than being a 'reward system' and the lack of differentiation of effects in the medial prefrontal cortex means that several other functions, probably including behavioral inhibition, are implicated. This is supported by the author's citation of their own original evidence on 'patience', although this is not a conventional way to describe behavioral function (and in my view, should be replaced). Note also that the sensorimotor cortex is activated (SS1 and M1); I see no reason why the "reward system" should specifically be highlighted.

2. It was disappointing not to have included data on behavioral correlates of these effects to justify the functional speculations.

3. It was also a pity that data on chronic effects of SSRIs could not have been included, to enhance possible clinical relevance. The authors have now cited some work on acute SSRI treatment, but this of course is not the most appropriate comparison with optogenetic activation, as it engages inhibitory autoreceptors; effects of chronic administration are much more relevant.

4. Minor "Foundational understanding" (line 35, abstract) is a somewhat hyperbolic phrase; I think just "understanding" would be sufficient. These findings are not fundamental.

The sentence "Nonetheless, how brain-wide serotonergic projections cause such effects and by which target areas and receptors, remain poorly understood [38, 64]." (lines 49-50 Introduction) may sound like a good rationale for this study, but, on reflection, it could be conjectured that this study has not actually successfully addressed this particular question. It may also under-represent previous work.

Overall, I accept that this study represents a good deal of excellent technical work and the findings of almost opposite effects on the BOLD response dependent on anaesthetised state is an interesting, novel finding, which helpfully resolves some confusion arising from a previous study published in this journal. This is the central contribution of the study as aptly summarised in the first paragraph of the Discussion. The significance or underlying causes of this opposite pattern are not readily apparent although possible mechanisms of effects of anaesthetics are speculated upon in the Discussion. The fact that the activation is related to DR topography and 5-HT receptor distributions is a re-assuring and useful validating finding, but surely not that surprising.

Reviewer #2 (Remarks to the Author):

This revised manuscript addresses some issues raised during the last review, but the authors did not perform any new experiments or data analyses for addressing reviewers' critiques. Mostly, speculative discussion and minor clarifications were made. This revised manuscript still has similar major issues mentioned previously.

Reviewer #3 (Remarks to the Author):

The authors have revised the manuscript in response to the earlier comments. The primary concern was the discrepancy between the main text and the figure legends. However, these issues have been addressed, and the reviewer was able to accurately compare the responses to both the

stimulation and control conditions in the revised manuscript.

Regarding my earlier comment for Figures 1g, 2c, 3d, S5, S6, and S11, the authors have clarified the stimulation protocol within the figure legends rather than visually representing the timing of blue and yellow light stimulation in the figures.

However, the reviewer found that the following figure explanations needed to be revised.

In Figure 4a, there is no color bar. The reviewer recommends that the authors add a color bar to the figure and provide an explanation within the figure legend.

Similarly, Figure 5a needs more information about the color bar within the figure legend. Please include a detailed description of what the color bar represents.

The same issue can be seen in Supplementary Figure S12; no information is provided for the color bar, mirroring the problem in Figure 5a.

Reviewer #4 (Remarks to the Author):

I think that the authors have adequately addresses my previous comments and I do not have any further concerns.

Reviewer #1 (Comments to the Author):

We thank the reviewer for the careful review and for suggesting further clarifications of our findings. Our responses are provided below in blue in point-by-point fashion.

1. I really think the emphasis on the reward system, for example even in the title of the ms, is inappropriate. The selection of caudate-putamen as part of the reward system is arbitrary, as Delgado amongst others clearly shows that the dorsal striatum can also be part of the 'punishment system'. It is perhaps also naive to describe the pallidum simply as part of the reward system. The hippocampus is implicated in appetitive functions as well as punishment. The frontal cortex clearly has other functions than being a 'reward system' and the lack of differentiation of effects in the medial prefrontal cortex means that several other functions, probably including behavioral inhibition, are implicated. This is supported by the author's citation of their own original evidence on 'patience', although this is not a conventional way to describe behavioral function (and in my view, should be replaced). Note also that the sensorimotor cortex is activated (SS1 and M1); I see no reason why the "reward system" should specifically be highlighted.

The serotonin system contributes to multiple functions, including reward and punishment processing. We agree that the emphasis on reward system in this study is misleading since activated brain regions, including the mPFC, striatum, and VTA contributed to the punishment system [16]. Instead, we used the term, valence system, to describe the selected brain regions.

Regarding patience, we instead rephrased this as waiting for delayed rewards, since waiting is a widely used term in the literature of impulsivity [17].

We removed the emphasis on reward circuits from the title and the text (Line 30-32, 64, 72, 106-111, and 243).

2. **It was disappointing not to have included data on behavioral correlates of these effects to justify the functional speculations.**

We agree that it is important to have behavioral correlates of brain responses to serotonin activations. We also performed behavioral experiments with 3 mice after fMRI experiments. However, we could not perform a statistically sufficient number of behavioral experiments. Instead, we added discussion of previous studies of behavioral correlates with serotonin activation.

A reduction of reward sensitivity is a symptom of major depression. In healthy subjects, 2-week SSRI administration potentiated reward signals in the ACC and vmPFC [18]. VTA-projecting DR serotonin neurons induce self-stimulating behaviors, indicative of positive reinforcement [12]. Brain responses from brain-wide cortical regions and VTA may reflect a mechanism underlying recovery of reward sensitivity. High delay discounting of future rewards is indicative of psychiatric disorders, including major depression [15]. A line of studies showed that optogenetic activations of DR serotonergic projections to the orbitofrontal cortex and mPFC, but not the NAcc, enhance waiting for delayed rewards [10-11]. Our brain response results support the concept that activations of OFC and mPFC are induced by optogenetic stimulation of DR serotonin neurons. High anxiety is a general symptom

in psychiatric disorders. SSRI-induced increases of extra-synaptic serotonin contributes to reduction of anxiety. However, simple activation of DR serotonin neurons did not induce antidepressant effects [14]. Antidepressant roles of DR serotonin neurons on anxiety can be region-dependent [5, 16] and work in a time-locked manner [13]. The frontal-cortex- and BST-projecting DR serotonin subsystems promote antidepressant effects whereas the amygdala-projecting DR serotonin subsystem enhances anxiogenic effects [5, 16]. Nishitani et al. (2018) showed that optogenetic serotonin neurons increase active coping with inescapable stress [13]. Our results support brain-wide involvement, including serotonergic activations of the frontal cortex and BST, which leads to antidepressant effects. However, such resting-state brain-wide activation also may blunt effects of co-activations of distinct subsystems, without time-locked stimulation.

We discussed behavioral correlates induced by serotonin activations (Line 246-268).

- 3. It was also a pity that data on chronic effects of SSRIs could not have been included, to enhance possible clinical relevance. The authors have now cited some work on acute SSRI treatment, but this of course is not the most appropriate comparison with optogenetic activation, as it engages inhibitory autoreceptors; effects of chronic administration are much more relevant.**

Serotonin is a primary therapeutic target for psychiatric disorders, especially with SSRI treatments. The reviewer noted that the therapeutic benefits of SSRI, including alleviation of depressed mood and psychic anxiety, arise from prolonged SSRI administration [4]. Chronic SSRI administration normalizes aberrant activity in the amygdala and ACC during major depressive episodes [3], as well as in brain networks associated with obsessive-compulsive disorder [7]. In healthy individuals, Scholl et al. (2017) found that chronic SSRI administration amplifies both reward and effort signals, fostering learning processes in the anterior cingulate cortex, amygdala, medial prefrontal cortex, and striatum [18]. Brain-wide positive responses from the ACC, mPFC, striatum, VTA with optogenetic activations in an awake state may partly imitate chronic SSRI administration.

Effects of chronic SSRI administration are discussed in Line 284-292.

4. Minor "Foundational understanding" (line 35, abstract) is a somewhat hyperbolic phrase; I think just "understanding" would be sufficient. These findings are not fundamental. The sentence "Nonetheless, how brain-wide serotonergic projections cause such effects and by which target areas and receptors, remain poorly understood [38, 64]." (lines 49-50 Introduction) may sound like a good rationale for this study, but, on reflection, it could be conjectured that this study has not actually successfully addressed this particular question. It may also under-represent previous work.

The main contribution of this work is to clarify which brain regions DRN serotonergic neurons regulate in the awake state and their association with structural density and molecular expression of 5-HT receptors. As the reviewer implied, our findings are not about the mechanism of brain-wide responses to DRN serotonin activations. Following the reviewer's comment, we removed the word 'foundational'

from Lines 35-37. We also specified the main target of this study to avoid over-estimation of our findings. In Lines 54-56, we specified which brain regions DRN serotonergic neurons regulate in an awake state and their association with structural density and molecular expression of 5-HT receptors.

Overall, I accept that this study represents a good deal of excellent technical work and the findings of almost opposite effects on the BOLD response dependent on anaesthetised state is an interesting, novel finding, which helpfully resolves some confusion arising from a previous study published in this journal. This is the central contribution of the study as aptly summarised in the first paragraph of the Discussion. The significance or underlying causes of of this opposite pattern are not readily apparent although possible mechanisms of effects of anaesthetics are speculated upon in the Discussion. The fact that the activation is related to DR topography and 5-HT receptor distributions is a re-assuring and useful validating finding, but surely not that surprising.

Reviewer #2 (Comments to the Author):

We thank the reviewer for suggesting improvements to our manuscript.

This revised manuscript addresses some issues raised during the last review, but the authors did not perform any new experiments or data analyses for addressing reviewers' critiques. Mostly, speculative discussion and minor clarifications were made. This revised manuscript still has similar major issues mentioned previously.

We agree that addressing the reviewer's critiques on the mechanism of negative BOLD response by DRN serotonin activation under anesthesia is an important research question; however, our research purpose was to show brain responses to DRN serotonin activation and their association with structural density and RNA maps of 5-HT receptors in an awake state. Hence, we do not think that it is necessary to perform further experiments to validate underlying mechanisms of the negative BOLD response. On the other hand, regarding the dissociation between projection density and the CBV response in the previous study, we will add further analyses.

We acknowledge two major points raised by Reviewer #2.

- 1. Our study did not explain/validate the underlying mechanism of negative BOLD caused by DRN serotonergic activation under an anesthetized state with isoflurane.**

Granjean et al. (2019) revealed that DRN serotonin activation induced a reduction in the amplitude of local field potentials (LFPs) and burst frequency with multiunit activity (MUA) from a variety of cortical regions, including M1, M2, S1, and FA under 0.5% isoflurane + 0.2 mg/kg/h s.c. medetomidine (Figure 3.a-d) [6]. That study also checked the association between brain responses by DRN serotonin activation and delta power/burst frequency (Figure 3.e-f) [6]. Our study demonstrated that a negative brain-wide response was caused by a reduction of cortical neuronal activity under isoflurane & medetomidine. Another study also showed that the negative BOLD response by activation of medial spiny neurons (MSNs) is associated with decreased neuronal

activity under isoflurane 0.3-0.7% (Figure 6H) [8]. These studies suggest that negative BOLD signals are induced by decreased neuronal activity.

There is also a possibility that the negative brain-wide response is due to an increase in cerebral blood flow (CBF), which subsequently reduces neuronal activity under isoflurane. It is well established that isoflurane affects blood vessels and amplifies CBF. Abe et al. (2021) showed that an increase in CBF in the striatum doesn't alter neuronal activity (Figure 7) [2]. Furthermore, increasing isoflurane dosages elevate BOLD signal levels (Figure 1A) [1]. Thus, these studies suggest that the decrease in BOLD signals isn't due to increased CBF. Overall, a brain-wide negative response can be induced by decreased neuronal activity rather than increased CBF.

We clarified the potential mechanism of negative BOLD signals under isoflurane in Lines 330-347.

- 2. Our study did not explain the dissociation between the projection density and the brain response under anesthesia in the previous study. Reviewer 2 suggested performing additional experiments with different anesthetics.**

There are several possible explanations for the dissociation between the projection density and the brain response under anesthesia, including scanner differences, genetic differences, and referred atlases. This is an important point, but we are not able to perform further experiments since no staff members are available for such experiments. Instead, we performed comparisons of brain responses between the current study and the Granjean dataset under our ROIs [6].

We used the open dataset (<https://doi.org/10.18112/openneuro.ds001541.v1.1.2>), and applied preprocessing with the concurrent study. Using a generalized linear model (GLM), we first constructed a functional map of CBV responses under anesthesia, based on stimulation parameters from a re-analysis of the original study [9]. As in the previous study, the functional map revealed a consistent brain-wide negative response ($p < 0.05$, FWE-corrected; Supplementary Figure 13). However, responses in the DRN also revealed a negative response, which was not seen in the previous study.

Supplementary Figure 13. Functional map of CBV responses to blue illumination (Grandjean et al., 2019) ($p < 0.05$, FWE-corrected; $n=8$, 63 runs).

We further checked whether the GLM may fail to capture characteristics of CBV responses in the DRN due to features of DRN responses. An average time series of CBV signals with 28 ROIs was extracted and visualized (Supplementary Figure 14a). The time series of CBV signals in the DRN revealed positive responses to optogenetic stimulation, consistent with the original study. This result indicates that comparisons between the CBV and BOLD signals with time series are preferred, since the GLM may fail to capture CBV responses in the DRN with the current stimulation parameters.

Supplementary Figure 14. Association of peak intensities between CBV signals [1, 3] and BOLD signals (Session3). (a) Time series of regional CBV signals from the mPFC, CPu, and DRN. Blue highlight indicates 20-sec stimulation cycles. (b) Association of peak intensities between CBV signals and BOLD signals under anesthesia (Session3). Statistically significant correlation between peak intensities between CBV signals and %BOLD signal was found ($r = 0.53$, $p < 0.01$). (c) Association between structural density and brain response. No statistically significant correlations of structural density with peak intensity of the CBV and %BOLD signals were found (vs. CBV signals (Grandjean et al., 2019): $r = 0.35$, $p = 0.08$; vs. %BOLD signals (Session 3): $r = 0.34$, $p = 0.09$). Deep and light magenta indicate the association of structural density with the peak of %BOLD and that of Δ CBV, respectively.

The peak intensity of time series of Δ CBV (Grandjean et al., 2019) and BOLD signals under anesthesia (Session 3) is further contrasted (Supplementary Figure 14b-c). We found a statistically significant positive correlation between peak intensity of Δ CBV and %BOLD signals under anesthesia ($r=0.53$, $p<0.01$; Supplementary Figure 14b). These results indicate that the difference between the previous and current studies originated from different selected ROI and methodological differences, including stimulation parameters of GLM.

We clarified analytical methods (Supplementary Methods, Line 57-103), and further discussed differences between our study and the previous study (Line 189-99, 414-5, 434-5, and 437-441; Supplementary Figure 13&14).

Reviewer #3 (Comments to the Author):

We thank the reviewer for careful checking of our figures and for making comments to improve reader understanding of our work.

The authors have revised the manuscript in response to the earlier comments. The primary concern was the discrepancy between the main text and the figure legends. However, these issues have been addressed, and the reviewer was able to accurately compare the responses to both the stimulation and control conditions in the revised manuscript.

Regarding my earlier comment for Figures 1g, 2c, 3d, S5, S6, and S11, the authors have clarified the stimulation protocol within the figure legends rather than visually representing the timing of blue and yellow light stimulation in the figures. However, the reviewer found that the following figure explanations needed to be revised.

- 1. In Figure 4a, there is no color bar. The reviewer recommends that the authors add a color bar to the figure and provide an explanation within the figure legend.**

The color bar at the left indicates beta values of BOLD responses corresponding to Figure 2a. The right color bar indicates normalized intensity of structural density. We added a color bar for Figure 4a and an explanation of it in Lines 743-745.

- 2. Similarly, Figure 5a needs more information about the color bar within the figure legend. Please include a detailed description of what the color bar represents.**

The color indicates scaled intensity of mRNA expression density of receptors and structural density for projections. The color bar ranges from 0-80. We added an explanation of the color bar in Figure 5a in Lines 764-65

- 3. The same issue can be seen in Supplementary Figure S12; no information is provided for the color bar, mirroring the problem in.**

Similar to Figure 5a, a bottom color bar indicates normalized intensity of mRNA expression density of receptors and structural density for projections. The color bar ranges from 0-80. We added an explanation of the color bar for Supplementary Figure S12 in Lines 320-321.

Reviewer #4 (Comments to the Author):

We appreciate the reviewer's positive comments on our manuscript.

I think that the authors have adequately addresses my previous comments and I do not have any further concerns.

We are very glad to hear that the manuscript poses no further concerns for the reviewer!

References

1. Abe, Y., T. Tsurugizawa, and D. Le Bihan, *Water diffusion closely reveals neural activity status in rat brain loci affected by anesthesia*. PLoS Biol, 2017. **15**(4): p. e2001494.
2. Abe, Y., et al., *Optical manipulation of local cerebral blood flow in the deep brain of freely moving mice*. Cell Rep, 2021. **36**(4): p. 109427.
3. Arnone, D., *Functional MRI findings, pharmacological treatment in major depression and clinical response*. Prog Neuropsychopharmacol Biol Psychiatry, 2019. **91**: p. 28–37.
4. Boschloo, L., et al., *The complex clinical response to selective serotonin reuptake inhibitors in depression: a network perspective*. Transl Psychiatry, 2023. **13**(1): p. 19.
5. Garcia-Garcia, A.L., et al., *Serotonin inputs to the dorsal BNST modulate anxiety in a 5-HT(1A) receptor-dependent manner*. Mol Psychiatry, 2018. **23**(10): p. 1990–1997.
6. Grandjean, J., et al., *A brain-wide functional map of the serotonergic responses to acute stress and fluoxetine*. Nat Commun, 2019. **10**(1): p. 350.
7. Kim, M., et al., *The effects of selective serotonin reuptake inhibitors on brain functional networks during goal-directed planning in obsessive-compulsive disorder*. Sci Rep, 2020. **10**(1): p. 20619.
8. Lee, H.J., et al., *Activation of Direct and Indirect Pathway Medium Spiny Neurons Drives Distinct Brain-wide Responses*. Neuron, 2016. **91**(2): p. 412–24.
9. Mandino, F., et al., *A triple-network organization for the mouse brain*. Mol Psychiatry, 2022. **27**(2): p. 865–872.
10. Miyazaki, K., et al., *Reward probability and timing uncertainty alter the effect of dorsal raphe serotonin neurons on patience*. Nat Commun, 2018. **9**(1): p. 2048.

11. Miyazaki, K., et al., *Serotonergic projections to the orbitofrontal and medial prefrontal cortices differentially modulate waiting for future rewards*. *Sci Adv*, 2020. **6**(48).
12. Nagai, Y., et al., *The Role of Dorsal Raphe Serotonin Neurons in the Balance between Reward and Aversion*. *Int J Mol Sci*, 2020. **21**(6).
13. Nishitani, N., et al., *Manipulation of dorsal raphe serotonergic neurons modulates active coping to inescapable stress and anxiety-related behaviors in mice and rats*. *Neuropsychopharmacology*, 2019. **44**(4): p. 721–732.
14. Ohmura, Y., et al., *Optogenetic activation of serotonergic neurons enhances anxiety-like behaviour in mice*. *Int J Neuropsychopharmacol*, 2014. **17**(11): p. 1777–83.
15. Pulcu, E., et al., *Temporal discounting in major depressive disorder*. *Psychol Med*, 2014. **44**(9): p. 1825–34.
16. Ren, J., et al., *Anatomically Defined and Functionally Distinct Dorsal Raphe Serotonin Sub-systems*. *Cell*, 2018. **175**(2): p. 472–487 e20.
17. Robinson, E.S., et al., *Behavioural characterisation of high impulsivity on the 5-choice serial reaction time task: specific deficits in 'waiting' versus 'stopping'*. *Behav Brain Res*, 2009. **196**(2): p. 310–6.
18. Scholl, J., et al., *Beyond negative valence: 2-week administration of a serotonergic antidepressant enhances both reward and effort learning signals*. *PLoS Biol*, 2017. **15**(2): p. e2000756.

REVIEWERS' COMMENTS

Reviewer #1 (Remarks to the Author):

The revised version is improved over the original and the authors have responded to my main comments, although the continued absence of actual relevant behavioral data and direct comparison of effects of SSRIs makes it a much less substantial study. The authors have relied on quite lengthy additions to an already lengthy Discussion in referring to relevant literature, which is inevitably (though understandably) somewhat biased to support their findings and is also suggesting at several points that further research is needed. The main contribution of the study in fact is to show in the mouse that activation of DRN serotonin neurons associated with BOLD responses in several terminal regions of the ramifying DRN projections and the likely involvement of post-synaptic 5HT1a receptors in these responses. This is perhaps not wholly surprising, although it helps to show that the previous finding published in Nat Comm under anaesthesia probably has less functional validity. However, the mechanistic basis for this state-dependent difference remains unexplained. Overall, the bold conclusion that "Our findings reveal a new physiological basis for understanding serotonergic control of brain-wide dynamics and functions" seems a little hard to justify.

Minor points:

1. valence is mis-spelled in one place (line 243)
2. Reference 17 is not the correct one for defining waiting or impulsivity. (It doesn't mention either term).
3. "brain-wide": I do not really understand the rationale of this description as it is somewhat misleading. It implies that very large numbers of brain regions are implicated in the DRN stimulation beyond those to which it actually projects and seems to imply an aspect of the findings which is surprising (but is in fact unwarranted). There were also no apparent ROIs that included important subcortical structures such as the hypothalamus. The authors targeted mainly the widely ramifying terminal regions of DRN- and that may be a more accurate description.
4. line 411-412 needs rephrasing. Perhaps "The neural mechanisms underlying the diverse ways in which serotonin modulates behavioral functions remain to be clarified"

Reviewer #4 (Remarks to the Author):

I do not have further comments.

Reviewer #1 (Comments to the Author):

We thank the reviewer for carefully reading our manuscript several times.

The revised version is improved over the original and the authors have responded to my main comments, although the continued absence of actual relevant behavioral data and direct comparison of effects of SSRIs makes it a much less substantial study. The authors have relied on quite lengthy additions to an already lengthy Discussion in referring to relevant literature, which is inevitably (though understandably) somewhat biased to support their findings and is also suggesting at several points that further research is needed. The main contribution of the study in fact is to show in the mouse that activation of DRN serotonin neurons associated with BOLD responses in several

terminal regions of the ramifying DRN projections and the likely involvement of post-synaptic 5HT1a receptors in these responses. This is perhaps not wholly surprising, although it helps to show that the previous finding published in Nat Comm under anaesthesia probably has less functional validity. However, the mechanistic basis for this state-dependent difference remains unexplained. Overall, the bold conclusion that "Our findings reveal a new physiological basis for understanding serotonergic control of brain-wide dynamics and functions" seems a little hard to justify.

1. valence is mis-spelled in one place (line 243)
We corrected the spelling of 'valence' (Line 246).
2. Reference 17 is not the correct one for defining waiting or impulsivity. (It doesn't mention either term).
Our original intent in referring to Robinson et al. in the previous response was to show that 'waiting' is used in the literature regarding 'impulsivity' [5].

[5] Robinson, E.S., et al., Behavioural characterisation of high impulsivity on the 5-choice serial reaction time task: specific deficits in 'waiting' versus 'stopping'. Behav Brain Res, 2009. 196(2): p. 310-6.

We cite here another paper showing that 'waiting' is clearly mentioned and defined in the impulsivity literature [6]. Accordingly, we believe that using the word 'waiting' is justifiable.

[6] Dalley, J.W., and Ersche, K.D. Neural circuitry and mechanisms of waiting impulsivity: relevance to addiction. Philos Trans R Soc Lond B Biol Sci, 2019. 374(1766).

3. "brain-wide": I do not really understand the rationale of this description as it is somewhat misleading. It implies that very large numbers of brain regions are implicated in the DRN stimulation beyond those to which it actually projects and seems to imply an aspect of the findings which is surprising (but is in fact unwarranted). There were also no apparent ROIs that included important subcortical structures such as the hypothalamus. The authors targeted mainly the widely ramifying terminal regions of DRN- and that may be a more accurate description.
The serotonin system, especially DRN, has widespread projections in the brain. We described 'brain-wide' activation as widespread activation from multiple brain areas, including cortical and subcortical areas, resulting from optogenetic DRN serotonin stimulation. Functional maps of optogenetic serotonin stimulation showed widespread activation, including cortical and subcortical areas (Figure. 2.a. & Supplementary Figure. S3). The previous study also described functional responses from optogenetic serotonin stimulation as, 'the brain-wide functional dynamics of the 5-HT projection system' [1]. Therefore, we believe that the use of "brain-wide" is reasonable.
4. line 411-412 needs rephrasing. Perhaps "The neural mechanisms underlying the diverse ways in which serotonin modulates behavioral functions remain to be clarified".
We rephrased the sentence as the reviewer suggested (Lines 410-411).

Once again, we thank the reviewer for his/her careful review.

Reviewer #4 (Comments to the Author):

We appreciate the reviewer's last comment on our manuscript.

I do not have further comments.

References

[1] Grandjean, J., et al., A brain-wide functional map of the serotonergic responses to acute stress and fluoxetine. Nat Commun, 2019. 10(1): p. 350.

- [2] Eklund, A., et al., BROCCOLI: Software for fast fMRI analysis on many-core CPUs and GPUs. *Front Neuroinform*, 2014. 8 (24).
- [3] Miyazaki, K., et al., Activation of Dorsal Raphe Serotonin Neurons Underlies Waiting for Delayed Rewards. *J Neurosci*, 2011. 31 (2): p. 469-79.
- [4] Puig M.V. et al., Serotonin modulates fast-spiking interneuron and synchronous activity in the rat prefrontal cortex through 5-HT1A and 5-HT2A receptors. *J Neurosci*, 30(6): p. 2211-22.
- [5] Robinson, E.S., et al., Behavioural characterisation of high impulsivity on the 5-choice serial reaction time task: specific deficits in 'waiting' versus 'stopping'. *Behav Brain Res*, 2009. 196(2): p. 310-6.
- [6] Dalley, J.W., and Ersche, K.D. Neural circuitry and mechanisms of waiting impulsivity: relevance to addiction. *Philos Trans R Soc Lond B Biol Sci*, 2019. 374(1766).